# Physiological febrile heat stress increases cytoadhesion through increased protein trafficking of *Plasmodium falciparum* surface proteins into the red blood cell

David Jones[1], Hugo Belda[1,2], Malgorzata Broncel[1], Gwendolin Fuchs[1,2], David Anaguano[2], Stephanie D Nofal[1,2], Moritz Treeck[1,2]*

[1]Signalling in Apicomplexan Parasites Laboratory, The Francis Crick Institute, London, United Kingdom; [2]The Cell Biology of Host-Pathogen Interactions Lab, Gulbenkian Institute for Molecular Medicine, Lisbon, Portugal

## eLife Assessment

This **important** study provides **compelling** evidence that fever-like temperatures enhance the export of *Plasmodium falciparum* transmembrane proteins, including the cytoadherence protein PfEMP1 and the nutrient channel PSAC, to the red blood cell surface, thereby increasing cytoadhesion. Using rigorous and well-controlled experiments, the authors **convincingly** demonstrate that this effect results from accelerated protein trafficking rather than changes in protein production or parasite development. These findings significantly advance our understanding of parasite virulence mechanisms and offer insights into how febrile episodes may exacerbate malaria severity.

*For correspondence: moritz.treeck@gimm.pt

**Abstract** Fever is a hallmark of malaria. Several studies have linked febrile temperatures to reduced parasite viability, but also to increased cytoadhesion, a key driver of pathology. However, different mechanisms have been proposed to cause changes in cytoadhesion and parasite sensitivity to heat. Here, we demonstrate that exposure of *Plasmodium falciparum*-infected red blood cells (iRBCs) to physiologically relevant febrile heat stress (39 °C), derived from patient data, enhances cytoadhesion through increased trafficking of the major virulence factor PfEMP1 to the iRBC surface. This phenomenon is not limited to PfEMP1 and common laboratory strains, as it extends to the surface nutrient channel PSAC in four clinical isolates of diverse geographic origin. The increased surface protein display occurs without changes in overall protein expression or parasite developmental progression. Using phosphoproteomics and proximity labelling, we find that elevated temperature also increases trafficking and phosphorylation of exported proteins into the RBC. Enhanced export is likely reliant on the presence of a transmembrane domain as shown by NanoLuc reporter assays. Collectively, our results indicate that febrile temperatures commonly experienced during infection can accelerate protein export, likely at the parasitophorous vacuole. This enhanced export following heat stress is relevant because increased cytoadhesion could influence disease severity through earlier iRBC sequestration and elevated bound parasite mass.

## Introduction

Malaria remains one of the most devastating infectious diseases and is caused by unicellular eukaryotic parasites of the *Plasmodium* genus. Of the six species that infect humans, *Plasmodium falciparum* is responsible for the vast majority of malaria-related deaths. An estimated 249 million malaria cases

**eLife digest** A fever usually indicates that the body is fighting an infection. In malaria, parasites infect red blood cells, and high temperature is a common symptom. Fever may slow parasite growth, but it may also increase levels of a parasite protein called PfEMP1 on the surface of infected cells. PfEMP1 makes these cells sticky, helping them attach to blood vessel walls and avoid removal by the spleen. This adhesion can worsen disease by blocking small blood vessels, including those in the brain. If the fever increases this stickiness, it could worsen symptoms. However, previous studies used to vary temperatures, leading to conflicting results.

Jones et al. tested whether a common fever temperature (39 °C) affects PfEMP1 levels and cell adhesion. Using infected human red blood cells, they found that 39 °C increases both adhesion to proteins found on blood vessel walls and the number of cells displaying PfEMP1 on their surface. To investigate the process behind this increase, they used techniques to measure the activity of proteins and their interactions with other human proteins. They showed that heat accelerates the export of certain parasite proteins to the red blood cell surface, including PfEMP1 and likely others, without increasing overall protein levels or parasite growth.

These findings suggest that fever may accelerate the surface export of certain parasite proteins, which unintentionally help infected cells adhere more strongly and earlier to blood vessels, potentially worsening the disease. However, further work is needed to confirm this in more realistic settings and to link it directly to disease severity in patients.

and 608,000 deaths occur annually (*WHO, 2023*). The pathology of malaria arises during the asexual replication phase of *P. falciparum* within RBCs. This cycle involves RBC invasion, intracellular growth, and the production of daughter merozoites, which are released upon host cell rupture, leading to exponential parasite growth.

Fever is a hallmark of malaria and a defining feature of the host immune response to *P. falciparum* infection. Malaria-associated fever is thought to arise from pathogen-associated and damage-associated molecular patterns (PAMPs and DAMPs) released during RBC rupture as parasites egress (*McGuire et al., 1998*; *Oakley et al., 2011*; *Kwiatkowski, 1995*). Febrile heat stress is frequently encountered by *P. falciparum*, and the parasite has likely evolved mechanisms to adapt to and prosper under these conditions. As asexual parasites progress through the 48 hr life cycle, they develop through three key stages: ring stage (0–24 hours post-invasion [hpi]), trophozoite stage (24–40 hpi), and schizont stage (40–48 hpi).

Previous studies found that the first half of the asexual life cycle (0–24 hpi) is more resistant to heat stress than the second half (24–48 hpi; *Kwiatkowski, 1989*). Research into the impact of elevated temperatures on *P. falciparum* growth and gene regulation has provided key insights into the parasite's response to heat stress. Random mutagenesis screening of 992 *P. falciparum* genes at 41 °C identified more than 200 mutants with significantly altered growth responses (*Zhang et al., 2021*). Transcriptomic analyses of wild-type *P. falciparum* following heat stress (40 °C and 41 °C) revealed significant changes in gene expression. *Oakley et al., 2007* found that these changes mirror apoptotic cell responses, while (*Tintó-Font et al., 2021*) identified a transcription factor (PfAP2-HS) required for parasite growth at both normal (37 °C) and elevated (40 °C) temperatures.

Cytoadhesion of *P. falciparum*-infected red blood cells (iRBCs) to the vascular endothelium is a key virulence trait. It is mediated by the exported and surface-displayed Erythrocyte Membrane Protein 1 (PfEMP1), which prevents splenic clearance. The specific PfEMP1 variant expressed determines the site of sequestration. For example, expression of the VAR2CSA variant enables iRBCs to bind chondroitin sulfate A (CSA) in the placenta (*Ayres Pereira et al., 2016*). PfEMP1-mediated sequestration in host organs is unique to *P. falciparum* and is a major driver of disease severity, contributing to serious complications such as cerebral and placental malaria (*Jensen et al., 2020*). Fever-like temperatures have been reported to increase PfEMP1 on the surface of RBCs, although this has been contested as described in more detail below (*Udomsangpetch et al., 2002*).

Approximately 10% of *P. falciparum*-encoded proteins are predicted to be exported beyond the parasite plasma membrane, with many thought to pass through the parasitophorous vacuole lumen (PVL) and parasitophorous vacuole membrane (PVM) into the RBC cytosol, although the exact routes

for transmembrane proteins are not fully understood (*Spielmann and Gilberger, 2015*; *Marti et al., 2004*; *de Koning-Ward et al., 2009*). These exported proteins remodel the host cell, mediating processes such as nutrient acquisition and immune evasion. Exported proteins contain one of two trafficking elements: PEXEL-containing proteins, which undergo N-terminal cleavage before export, and PEXEL-negative exported proteins (PNEPs) which are not cleaved but contain transmembrane domains required for export (*Marti et al., 2004*; *Hiller et al., 2004*; *Spielmann and Gilberger, 2010*).

PfEMP1 is trafficked to the RBC surface via parasite-derived membranous organelles known as Maurer's clefts (*Cowman et al., 2016*; *Carmo et al., 2022*; *Maier et al., 2008*; *Maier et al., 2007*). This process relies on several exported proteins, many unique to *P. falciparum*, including FIKK4.1, an exported kinase that localises to the RBC periphery (*Davies et al., 2020*). Deletion of FIKK4.1 leads to a 55% reduction in PfEMP1-mediated cytoadhesion (*Davies et al., 2020*). Even though several proteins have been identified that are important for PfEMP1 trafficking, the precise mechanisms that underlie PfEMP1 surface translocation remain poorly understood (*Maier et al., 2008*; *Chan et al., 2016*).

Several studies have examined the effects of febrile heat stress on iRBC cytoadhesion, showing contradictory results. *Udomsangpetch et al., 2002* reported that heat stress increased iRBC cytoadhesion to endothelium expressing the PfEMP1 cognate receptor, correlating with higher PfEMP1 levels on the iRBC surface. In contrast, *Oakley et al., 2007* found no change in the proportion of iRBCs displaying PfEMP1 on the surface following heat stress. Furthermore, trophozoite stage iRBCs become more rigid under heat stress, a property predicted to aid parasite sequestration in organs (*Marinkovic et al., 2009*). *Zhang et al., 2018* reported an alternative mechanism whereby cytoadhesion increased due to elevated phosphatidylserine (PS) and not PfEMP1 on the iRBC surface following heat stress. One protein that has been linked to both heat stress and PfEMP1 surface levels is the *P. falciparum* exported protein HSP70x. *Charnaud et al., 2017*, but not *Cobb et al., 2017*, found HSP70x to be important for PfEMP1 trafficking, although both studies concluded that HSP70x is dispensable for intraerythrocytic parasite growth at 37 °C. *Charnaud et al., 2017*, further reported no significant growth differences in HSP70x knockout parasites under heat stress or nutrient-limiting conditions. In contrast, *Day et al., 2019* observed reduced parasitaemia in HSP70x knockdown parasites in the cycle following heat stress, compared to wild-type strains. In summary, whether febrile temperatures modulate cytoadhesion and by what mechanism remains unclear. Given that increased cytoadherent iRBC biomass in febrile patients could contribute to disease severity, further studies are important.

The divergent outcomes of the above-mentioned studies could be due to differences in the heat-stress conditions used, including those that affect parasite viability. Furthermore, in vitro culture of *P. falciparum* can cause gene loss of exported proteins, and transfections can impose a genetic bottleneck that may enrich such mutants (*Kemp et al., 1992*). It is therefore not uncommon to observe defects in cytoadhesion after generating transgenic parasites. Consequently, loss of cytoadhesion without restoring the phenotype by gene complementation or using conditional approaches should be interpreted with caution.

Here, we assessed how febrile heat stress influences exported protein trafficking and cytoadhesion in *P. falciparum*. We first defined three key criteria: (1) a physiologically relevant heat stress commonly occurring in malaria patients, (2) its impact on parasite growth across asexual stages, and (3) the normal timing of PfEMP1 trafficking to the iRBC surface. Using this framework, we tested whether febrile heat stress modulates iRBC cytoadhesion and its underlying mechanism. We show that heat stress increases cytoadhesion of iRBCs, coinciding with a greater proportion of infected cells displaying PfEMP1 on their surface. Comparative phosphoproteomic analysis and proximity labelling of the Maurer's clefts protein environment suggest that elevated temperatures enhance protein trafficking into the RBC cytosol. NanoLuciferase (NanoLuc) fusion proteins and compartment-specific isolation confirmed a greater abundance of PfEMP1 in the RBC cytosol following heat stress. Constitutively expressed reporter proteins suggest that transmembrane domains are key contributors to this increased trafficking, independent of overall protein expression levels. This increased surface trafficking is not limited to PfEMP1. It also extends to the Plasmodial Surface Anion Channel (PSAC), a parasite-derived nutrient channel, across four *P. falciparum* clinical isolates from diverse geographic origins.

# Results

## A physiologically relevant and non-destructive febrile heat stress increases iRBC cytoadhesion and surface PfEMP1 display

While a variety of temperatures mimicking febrile heat stress have been used in previous studies, these have generally not been assessed in relation to parasite viability, or common febrile temperatures in patients. Therefore, we sought to initially establish conditions that reflect the typical febrile response in malaria patients, focusing on a time window when PfEMP1 is exported and parasite viability remains unaffected.

To approximate the most common febrile temperature in individuals infected with *P. falciparum*, we analysed patient temperature data from two publicly available supplementary datasets (*Mok et al., 2015*; *Abdi et al., 2023*). The average tympanic febrile temperature (defined as greater than 38 °C) in *P. falciparum* malaria cases was 38.9 °C and 39.1 °C, respectively (*Figure 1A–B*). Temperatures ≥ 41 °C were rarely recorded in either dataset. Analysis of microarray data from *Mok et al., 2015* showed that higher patient temperatures were associated with a younger average age of circulating parasites, possibly reflecting increased cytoadherence and sequestration of more mature stages during fever (*Figure 1C*).

To assess the impact of heat stress on *P. falciparum* (NF54 DiCre strain; *Tibúrcio et al., 2019*) growth and replication, parasites were exposed to elevated temperatures (38–41 °C) for 8 hr across six different developmental time windows. Parasitaemia was measured in the subsequent cycle (*Figure 1D*). Similar to the observations by *Kwiatkowski, 1989*, we found that parasites in the first half of the cycle (0–24 hpi) were more resistant to heat stress than those in the second half (24–48 hpi). However, our data further reveal that recently invaded parasites (0–8 hpi) are less resistant to elevated temperatures compared to later ring-stage parasites (8–16 hpi and 16–24 hpi). We also observe that exposure to 41 °C for 8 hr is highly destructive to the parasite, irrespective of the developmental stage at which it is applied.

To test whether PfEMP1 surface translocation is affected at the mean febrile temperature observed in patients, we first determined the percentage of iRBCs maintained at 37 °C that display PfEMP1 (VAR2CSA) on the surface from 12 to 40 hpi. These experiments revealed a dynamic window of surface expression between 16 and 32 hpi, consistent with previous reports (*Looker et al., 2019*; *Knuepfer et al., 2005*; *Figure 1E*). These foundational data informed the identification of a heat stress window (39 °C) that mimics fever-range temperatures, overlaps with VAR2CSA trafficking to the iRBC surface (16–24 hpi), and was not expected to impair parasite growth.

We next subjected tightly synchronised *P. falciparum* cultures to elevated temperatures (39 °C) or control conditions (37 °C; *Figure 1F*) to assess the impact of heat on cytoadhesion and PfEMP1 surface translocation. Following heat stress, significantly more iRBCs (57.6%±19.4% more than non-heat stressed controls) cytoadhered under physiological flow conditions (1 dyne/cm²) compared to those maintained at 37 °C, which was set as the baseline and normalised to 100% (*Figure 1G*). This increase in cytoadhesion correlated with a significantly higher percentage of iRBCs displaying PfEMP1 (VAR2CSA) on their surface (67.4% ± 3.93% vs 46.9%±6.18%) compared to the control (*Figure 1H*). Importantly, heat stress between 16 and 24 hpi did not accelerate parasite development through the asexual life cycle (*Figure 1—figure supplement 1*).

Collectively, these data, in line with *Udomsangpetch et al., 2002* show that more iRBCs cytoadhere following febrile heat stress as the parasites develop from ring stages to trophozoites. This coincides with a significantly greater number of iRBCs displaying PfEMP1 on the surface. The mechanism underlying the increased translocation of PfEMP1 to the surface remains unknown.

## RBCs infected with four *P. falciparum* strains from different geographic origins exhibit increased sensitivity to sorbitol following heat stress

To investigate whether proteins other than PfEMP1 are trafficked differently during heat stress, we sought to identify another cell surface protein that is both detectable and essential for in vitro growth, ensuring its expression across all *P. falciparum* strains. One such protein complex is the Plasmodial Surface Anion Channel (PSAC), which facilitates nutrient uptake and is exported to the surface of iRBCs. PSAC comprises three proteins: RhopH2, RhopH3, and one of several cytoadherence-linked asexual gene (CLAG) variants (*Ekland et al., 2011*; *Pasternak et al., 2022*), which contain a putative

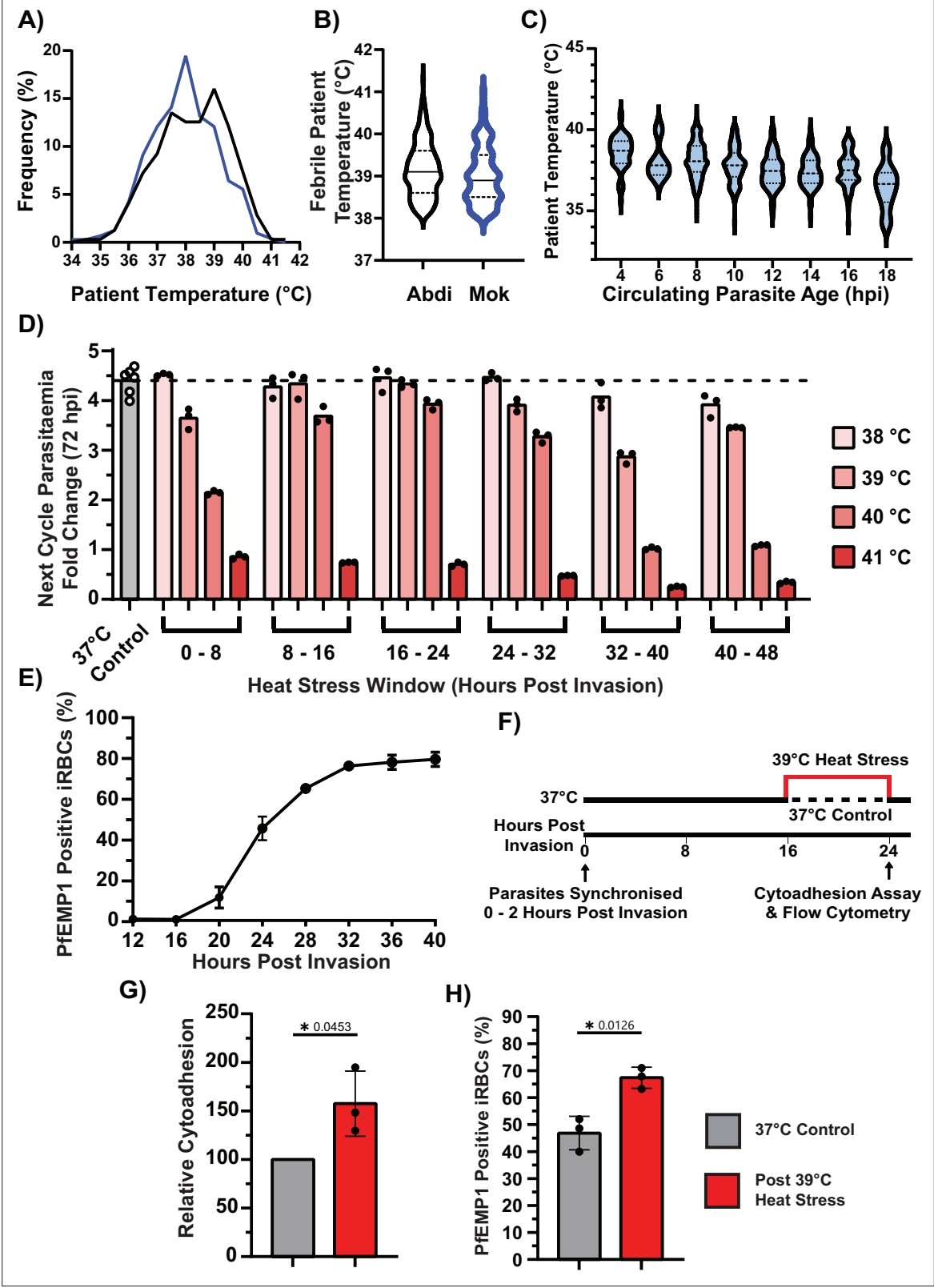

**Figure 1.** A physiologically relevant, non-destructive febrile heat stress increases the percentage of *P. falciparum* infected red blood cells (iRBCs) expressing PfEMP1 on their surface and enhances cytoadhesion to the cognate receptor. (**A**) Relative distribution of patient body temperature from two studies (*Mok et al., 2015* and *Abdi et al., 2023*) that included accessible temperature data from *P. falciparum* infected patients (n=1043 [blue] and n=827 [black], respectively). (**B**) Average febrile temperature, defined as >38 °C, was 38.9 °C and 39.1 °C in Mok et al., and Abdi et al., respectively.

*Figure 1 continued on next page*

*Figure 1 continued*

(**C**) Estimated average age of circulating parasites from Mok et al. microarray data plotted against recorded patient temperature. (**D**) Synchronised iRBCs were exposed to heat stress for different 8 hr windows during the first 48 hours post-invasion (hpi). When not heat-stressed, parasites were cultured under normal conditions at 37 °C. Parasitaemia was measured by flow cytometry in the following cycle at 72 hpi. N=3 (technical triplicates) for heat-stressed cultures; n=6 (biological replicates) for 37 °C control cultures. Dashed lines represent the average next cycle parasitaemia fold change of control cultures. (**E**) Percentage of iRBC positive for PfEMP1 (VAR2CSA) surface expression from 12 to 40 hpi when cultured normally at 37 °C. Flow cytometry gating strategy is shown in *Figure 1—figure supplement 2*. (**F**) Schematic representation of the heat stress application to synchronous NF54 DiCre parasites. (**G**) Following heat stress, more iRBCs bound to CSA under flow (1 dyne/cm²). (**H**) Significantly more iRBCs were positive for PfEMP1 (VAR2CSA) after heat stress. For G and H, n=3 biological replicates. Error bars displayed are ±1 SD. Statistical significance was determined using unpaired t-tests of log-transformed data with Welch's correction. Flow cytometry and cytoadhesion assays were performed at room temperature.

The online version of this article includes the following source data and figure supplement(s) for figure 1:

**Source data 1.** Total patient temperature data from Abdi et al. and Mok et al.

**Source data 2.** Only feverish (>38 °C) patient temperature data from Abdi et al. and Mok et al.

**Source data 3.** Patient data from Mok et al., correlating the temperature (°C) with the estimated parasite age (hpi).

**Source data 4.** Raw data file of next cycle parasitaemia fold change following different temperature conditions.

**Source data 5.** Raw data file of infected RBCs positive for cell surface PfEMP1 over 40 hours post-invasion when cultured at 37 °C.

**Source data 6.** Raw data file of the relative cytoadhesion of iRBCs following heat stress compared to iRBCs maintained at 37 °C.

**Source data 7.** Raw data file of the percentage of iRBC positive for cell surface PfEMP1 (VAR2CSA) following heat stress compared to iRBCs maintained at 37 °C.

**Figure supplement 1.** Heat stress applied between 16 and 24 hours post-invasion (hpi) does not accelerate *P. falciparum* asexual development.

**Figure supplement 1—source data 1.** Raw data file of infected RBCs Hoechst (DNA stain) MFI following heat stress or when maintained at 37 °C.

**Figure supplement 1—source data 2.** Raw data file of percentage of infected RBCs which had egressed at 40 hours post-invasion following heat stress or when maintained at 37 °C.

**Figure supplement 1—source data 3.** Raw data file of the fold change in parasitaemia in the next cycle (72 hpi) following heat stress or when maintained at 37 °C.

**Figure supplement 2.** Flow cytometry gating strategy for detection of surface-trafficked VAR2CSA on iRBCs.

**Figure supplement 3.** Temperature probe within cell culture flasks accurately measures heat stress conditions applied to cell cultures.

**Figure supplement 3—source data 1.** Raw data file of the temperature in a T25 tissue culture flask that has been heat stressed by transferring the flask from a 37 °C incubator to a 39 °C incubator.

transmembrane domain (TMD; *Sharma et al., 2015*; *Ho et al., 2021*; *Trickey et al., 2025*). All three components are expressed during the late stages of the parasite life cycle. These pre-formed proteins are stored in the parasitophorous vacuole membrane (PVM) following invasion and are trafficked into the host cell via the *Plasmodium* translocon of exported proteins (PTEX) complex at a time similar to PfEMP1 (*de Koning-Ward et al., 2009*; *Ito et al., 2017*; *Shao et al., 2022*).

Unlike PfEMP1, PSAC-mediated nutrient uptake is essential for parasite survival under in vitro conditions (*Kirk et al., 1994*; *Desai et al., 2000*; *Pillai et al., 2012*). Functional PSAC on the iRBC surface transports the small molecule sorbitol into the RBC cytosol, disrupting the osmotic balance of the cell and rupturing the iRBC (*Figure 2A–B*). Consequently, the percentage of ruptured iRBCs following sorbitol treatment serves as a surrogate marker for functional PSAC present at the iRBC surface (*Desai et al., 2005*).

We found that significantly more NF54 iRBCs were sensitive to sorbitol treatment following heat stress (53.3% ± 5.2%) compared to normal conditions (26.2% ± 8.6%; *Figure 2C–D*). Sorbitol lysis affected only iRBCs and was blocked by the PSAC inhibitor furosemide, showing that heat stress does not impact uninfected RBC (uRBC) integrity or induce PSAC-independent sorbitol uptake (*Figure 2—figure supplement 1*). To assess whether heat-induced changes in protein surface trans-location are conserved across *P. falciparum* strains, we extended our analysis to three geographically distinct isolates: HB3 (Isolated in Honduras 1984), Cam3.II (Isolated in Cambodia 2012), and HL2208 (Isolated in Uganda 2024; *Delemarre-van de Waal and de Waal, 1981*; *Walliker et al., 1987*; *Amaratunga et al., 2012*; *van Schalkwyk et al., 2024*). In all three strains, heat stress (39 °C) resulted in a significant increase in iRBC sensitivity to sorbitol lysis (*Figure 2E–G*), consistent with findings in the NF54 strain. Unless PSAC components are modified (e.g. through post-translational modifications) to increase PSAC activity following heat stress, these results indicate that earlier trafficking of PSAC to

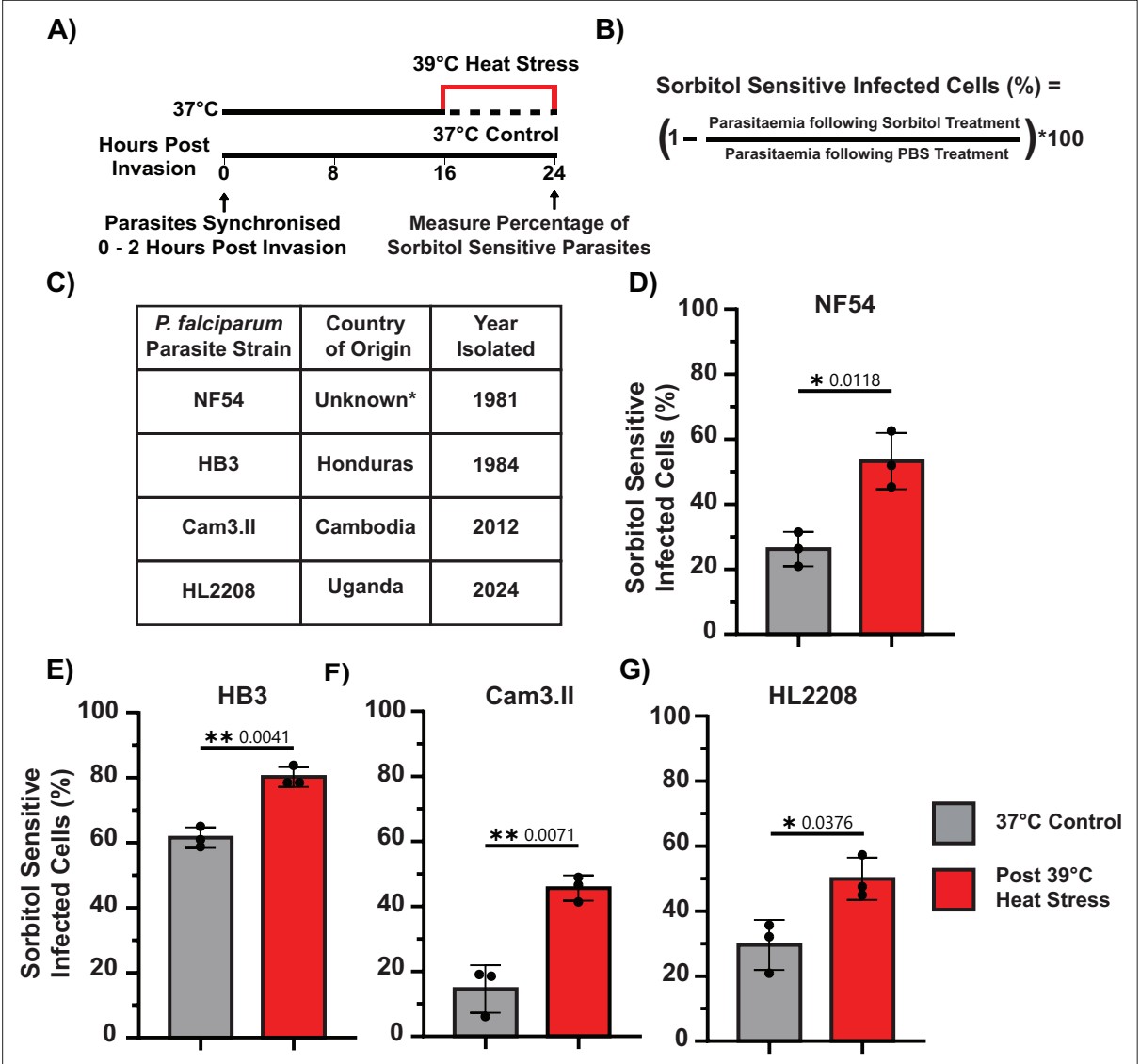

**Figure 2.** Heat stress increases sorbitol sensitivity in RBCs infected with four *P. falciparum* strains from diverse geographic origins. (**A**) At 24 hpi following an 8 hr heat stress (39 °C), parasite cultures were treated with PBS and sorbitol, and the parasitaemia was determined and compared to that of a 37 °C control. Sorbitol sensitivity is conferred by a functional nutrient channel (PSAC) trafficked by the parasite onto the surface of iRBCs. (**B**) As sorbitol-sensitive parasites rupture during treatment, the percentage of sorbitol-sensitive parasites is calculated by comparing parasitaemia following sorbitol treatment to parasitaemia following PBS treatment. (**C**) Geographic origins of *P. falciparum* isolates (*NF54 is predicted to have an African origin but was isolated in the Netherlands *Preston et al., 2014*). (**D–G**) After heat stress, NF54 (**D**), HB3 (**E**), Cam3.II (**F**), and HL2208 (**G**) strains exhibited significantly higher sensitivity to sorbitol treatment. Data represents three biological replicates (N=3). Statistical significance was assessed using unpaired t-tests on log-transformed data with Welch's correction. All assays were performed at room temperature.

The online version of this article includes the following source data and figure supplement(s) for figure 2:

**Source data 1.** Raw data file of iRBCs (NF54 strain) sorbitol sensitivity following heat stress compared to when maintained at 37 °C.

**Source data 2.** Raw data file of iRBCs (HB3 strain) sorbitol sensitivity following heat stress compared to when maintained at 37 °C.

**Source data 3.** Raw data file of iRBCs (Cam3.II strain) sorbitol sensitivity following heat stress compared to when maintained at 37 °C.

**Source data 4.** Raw data file of iRBCs (HL2208 strain) sorbitol sensitivity following heat stress compared to when maintained at 37 °C.

**Figure supplement 1.** Sorbitol lysis of infected red blood cells (iRBCs) is reduced by the PSAC inhibitor furosemide regardless of temperature treatment.

**Figure supplement 1—source data 1.** Raw data file of the cell supernatant absorbance (415 nm) following the different treatment of uninfected and infected RBCs.

**Figure supplement 2.** *P. falciparum* parasites are not more susceptible to heat stress with reduced nutrient availability.

*Figure 2 continued on next page*

*Figure 2 continued*

**Figure supplement 2—source data 1.** Raw data showing the percentage of infected red blood cells under normal or reduced nutrient conditions with defined media composition or nutrient channel inhibitors, following heat stress or when maintained at 37 °C.

the iRBC surface under febrile heat stress is a conserved phenomenon across diverse *P. falciparum* isolates. Culturing parasites in sub-lethal furosemide concentrations or in reduced nutrient media leads to reduced parasitaemia (*Figure 2—figure supplement 2*). However, the parasitaemia is not further reduced following heat stress. This shows that increased PSAC levels/activity do not enhance parasite survival under conditions of limited nutrient availability either from furosemide-induced nutrient deprivation or a reduced nutrient media composition. Collectively, these results indicate that the increased membrane trafficking described above is not restricted to PfEMP1.

## A *P. falciparum* heat stress phosphoproteome reveals substantial enrichment in phosphorylation of exported proteins by exported FIKK kinases

We next sought to identify factors that influence protein trafficking to the iRBC surface. Our previous work identified the exported protein kinase FIKK4.1 as important for efficient surface translocation of PfEMP1 (*Davies et al., 2020*). This kinase is also required for phosphorylation of RhopH3 (PF3D7_0905400), a component of the PSAC complex (*Davies et al., 2020*; *Davies et al., 2023*) and may therefore play a role in regulating the trafficking of both proteins. FIKK4.1 is part of the FIKK kinase family, which includes 19 kinases in *P. falciparum*, 18 of which are likely exported into the RBC. Several other exported kinases, including FIKK10.2, a kinase localised to the Maurer's clefts, phosphorylate proteins involved in PfEMP1 export but do not affect PfEMP1 trafficking at 37 °C (*Davies et al., 2023*; *Ward et al., 2004*). To broadly investigate whether FIKK kinases respond to elevated temperatures and potentially contribute to increased surface translocation of PfEMP1 and PSAC, we measured the phosphoproteome of iRBCs following heat stress and control conditions.

Following the same heat stress window previously shown to increase cytoadhesion (16–24 hpi, 39 °C), parasites were returned to 37 °C until 30 hpi. At this point, iRBCs were enriched from uninfected cells by density separation and lysed in 8 M urea. Proteins were digested and labelled with tandem mass tags (TMT), and enriched phosphopeptides were identified by LC-MS/MS (*Figure 3A*, *Supplementary file 1 and 2*). A total of 196 phosphopeptides (1.2% of those detected) were differentially phosphorylated following heat stress, with 143 showing increased phosphorylation (>1 Log2FC) and 53 showing decreased phosphorylation (<-1 Log2FC). Notably, 71% of the heat-enriched phosphopeptides mapped to exported proteins, which far exceeds the approximately 10% of the total proteome predicted to be exported (*Figure 3B–C*; *Spielmann and Gilberger, 2015*; *Marti et al., 2004*). Analysis of non-phosphorylated peptide levels showed no major changes in either the human or parasite proteomes following heat stress (*Figure 3—figure supplement 1*). This indicates that neither parasite development nor total protein expression accounts for the observed increase in surface PfEMP1.

We have previously mapped the phosphorylation events that depend on each FIKK kinase under normal temperature (37 °C) conditions (*Davies et al., 2020*). To determine whether the heat-induced phosphosites were novel or dependent on known FIKKs, we compared heat-induced phosphosites with our prior datasets (*Davies et al., 2020*). This analysis suggested that most heat-dependent phosphorylation could be largely attributed to FIKK10.2 and, to a lesser extent, FIKK4.1 (*Figure 3D*). To validate these results, we performed a second phosphoproteomic analysis following heat stress conditions using several previously generated exported FIKK knockout strains (FIKK1, FIKK4.1, FIKK4.2, all FIKK9s, FIKK10.1, and FIKK10.2). This confirmed that FIKK10.2 is required for the majority of phosphorylation increases observed in exported proteins during heat stress (*Figure 3—figure supplement 2*; *Supplementary file 3*).

Interestingly, while FIKK4.1 was previously shown to be necessary for efficient PfEMP1 surface translocation at 37 °C (*Davies et al., 2020*), we still observed a significant increase in PfEMP1 surface levels in the absence of FIKK4.1 following heat stress. Despite being critical for most heat-stress phosphorylation events, FIKK10.2 was also not required for the heat-induced increase in PfEMP1 surface presentation or sorbitol sensitivity (*Figure 3E–F*).

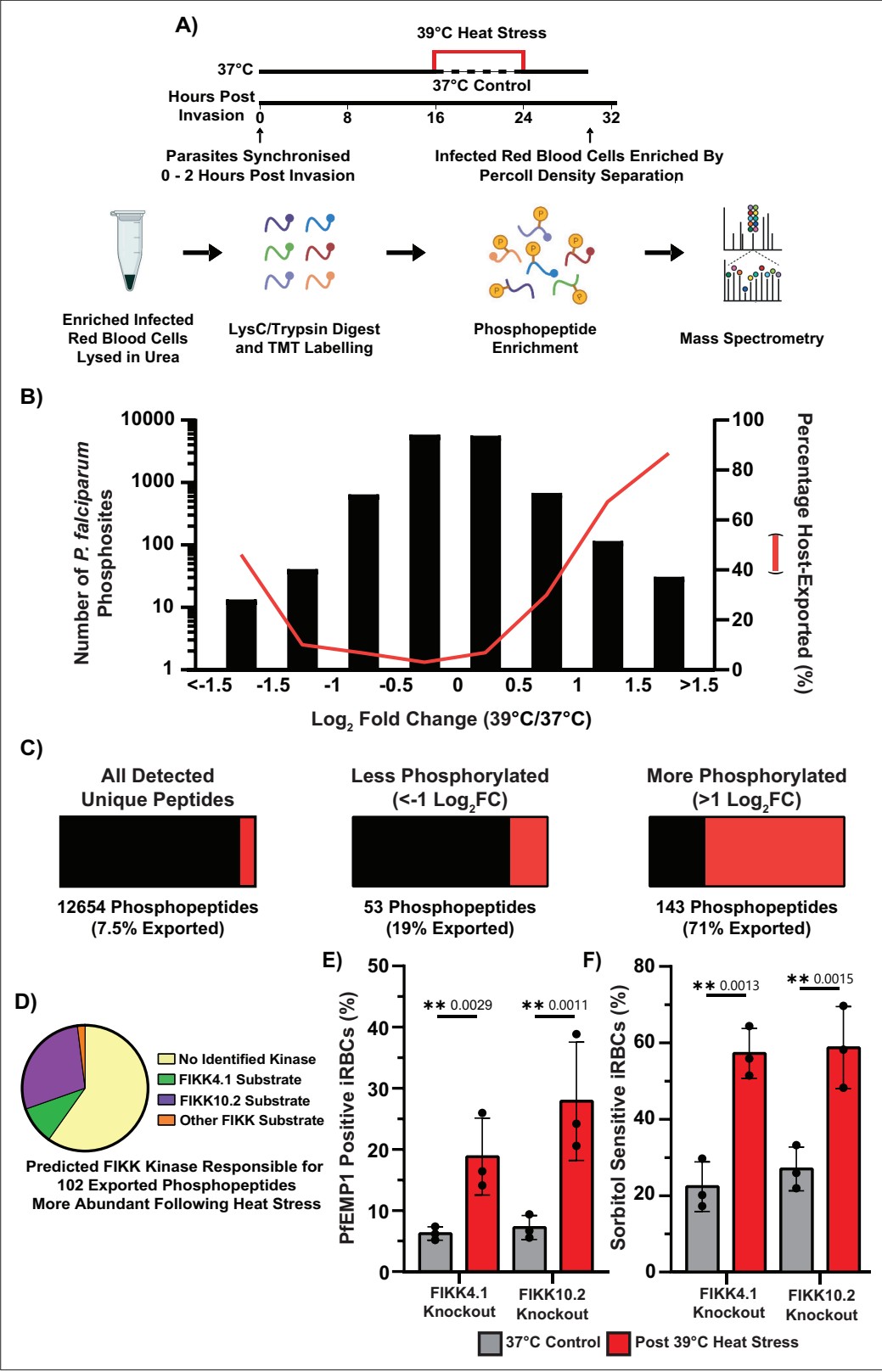

**Figure 3.** Febrile heat stress results in increased phosphorylation of host cell exported proteins. (**A**) Overview of infected RBC lysate preparation for phosphoproteomics. (**B**) Histogram of 12,654 unique *P. falciparum* phosphopeptides, showing enrichment of phosphopeptides from exported parasite proteins following heat stress. (**C**) Percentage of unique phosphopeptides that are from parasite exported proteins (red). Following heat

*Figure 3 continued on next page*

*Figure 3 continued*

stress, the significantly more abundant phosphopeptides (>1 Log2FC) are enriched for exported proteins (71%) compared to the 12,564 phosphopeptides not changing (7.5%). N=2 biological replicates. (**D**) Comparing the 102 exported phosphopeptides more abundant following heat stress against confident FIKK substrates identified FIKK10.2 and FIKK4.1 as likely major contributors to heat stress phosphorylation. The remaining phosphopeptides had not previously been linked to a specific kinase (No Identified Kinase) by *Davies et al., 2020*. Neither FIKK4.1 nor FIKK10.2 is required for the heat stress-induced increase in PfEMP1 surface expression (**E**) or for the increased sorbitol sensitivity of iRBCs following heat stress (**F**), as shown using synchronised kinase conditional knockout (RAP-treated) parasite lines. N=3 biological replicates. Error bars displayed are ±1 SD. Statistical significance was determined using the one-way ANOVA test with the Benjamini, Krieger, and Yekutieli FDR correction. Antibody staining, flow cytometry, and sorbitol treatment were performed at room temperature. For (**E**) and (**F**), parasites were heat stressed between 16 and 24 hpi at 39 °C, sorbitol treatment and antibody staining were performed at 24 hpi.

The online version of this article includes the following source data and figure supplement(s) for figure 3:

**Source data 1.** Raw data file of the distribution of *P. falciparum* phosphopeptide abundance $Log_2$ fold change following heat stress compared to when maintained at 37 °C.

**Source data 2.** Raw data file of the four categories of previously identified FIKK substrates of the *P. falciparum* phosphopeptides more abundant following heat stress.

**Source data 3.** Raw data file of the percentage of FIKK4.1 and FIKK10.2 knockout iRBCs positive for cell surface PfEMP1 (VAR2CSA) following heat stress compared to iRBCs maintained at 37 °C.

**Source data 4.** Raw data file of the percentage of FIKK4.1 and FIKK10.2 knockout iRBCs which are sensitive to sorbitol following heat stress compared to iRBCs maintained at 37 °C.

**Figure supplement 1.** The host and parasite proteomes are not substantially altered following febrile heat stress.

**Figure supplement 1—source data 1.** Raw data file containing normalised protein abundance values of parasite proteins under control conditions and following febrile heat stress.

**Figure supplement 1—source data 2.** Raw data file containing normalised protein abundance values for host and parasite proteins under control conditions and following febrile heat stress.

**Figure supplement 2.** Repeated heat stress phosphoproteome with FIKK knockout (KO) strains identifies FIKK10.2 as the major contributor of heat-stress-dependent phosphorylation of host-cell exported parasite proteins.

**Figure supplement 2—source data 1.** Raw data file containing abundance values for exported phosphopeptides that were significantly more abundant following heat stress in wild-type NF54 DiCre parasites, with corresponding values for the FIKK kinase knockout strains under control and heat stress conditions.

**Figure supplement 3.** Generation of HSP70x-3xHA (PF3D7_0831700) conditional knockout *P. falciparum* strain.

**Figure supplement 3—source data 1.** Raw images of the DNA gel assessing the correct integration of the HSP70x-3xHA transgenic parasite strain, the efficiency of the DiCre-mediated conditional knockout and the absence of the parental strain.

**Figure supplement 3—source data 2.** Raw images of the α-HA immunoblot of HSP70x-3xHA lysates.

**Figure supplement 3—source data 3.** Raw images of the total protein content control of the α-HA immunoblot of HSP70x-3xHA lysates.

**Figure supplement 3—source data 4.** PDF containing the raw images of the DNA gel assessing the correct integration of the HSP70x-3xHA transgenic parasite strain, the efficiency of the DiCre-mediated conditional knockout and the absence of the parental strain.

**Figure supplement 3—source data 5.** PDF containing the raw images of the α-HA immunoblot of HSP70x-3xHA lysates.

**Figure supplement 3—source data 6.** PDF containing the raw images of the total protein content control of the α-HA immunoblot of HSP70x-3xHA lysates.

**Figure supplement 4.** HSP70x-3xHA (PF3D7_0831700) does not strongly co-localise with KAHRP or SBP1.

**Figure supplement 4—source data 1.** Raw images.

**Figure supplement 4—source data 2.** Raw images.

Raw microscopy images of transgenic HSP70x-3xHA parasites probed with α-HA and α-KAHRP.

**Figure supplement 4—source data 3.** PDF containing the raw microscopy images of transgenic HSP70x-3xHA parasites probed with α-HA and α-SBP1.

**Figure supplement 4—source data 4.** PDF containing the raw microscopy images of transgenic HSP70x-3xHA

*Figure 3 continued on next page*

*Figure 3 continued*

parasites probed with α-HA and α-KAHRP.

**Figure supplement 5.** Generation and validation of PF3D7_0702500-3xHA conditional knockout *P. falciparum* strain.

**Figure supplement 5—source data 1.** Raw images of the DNA gel assessing the correct integration of the PF3D7_0702500-3xHA transgenic parasite strain, the efficiency of the DiCre-mediated conditional knockout and the absence of the parental strain.

**Figure supplement 5—source data 2.** Raw images of the α-HA immunoblot of PF3D7_0702500-3xHA lysates.

**Figure supplement 5—source data 3.** Raw images of the total protein content control of the α-HA immunoblot of PF3D7_0702500-3xHA lysates.

**Figure supplement 5—source data 4.** PDF containing the raw images of the DNA gel assessing the correct integration of the PF3D7_0702500-3xHA transgenic parasite strain, the efficiency of the DiCre-mediated conditional knockout and the absence of the parental strain.

**Figure supplement 5—source data 5.** PDF containing the raw images of the α-HA immunoblot of PF3D7_0702500-3xHA lysates.

**Figure supplement 5—source data 6.** PDF containing the raw images of the total protein content control of the α-HA immunoblot of PF3D7_0702500-3xHA lysates.

**Figure supplement 6.** PF3D7_0702500-3xHA co-localises with SBP1.

**Figure supplement 6—source data 1.** Raw images.

**Figure supplement 6—source data 2.** Raw images.

**Figure supplement 6—source data 3.** PDF containing the raw microscopy images of transgenic PF3D7_0702500-3xHA parasites probed with α-HA and α-SBP1.

**Figure supplement 6—source data 4.** PDF containing the raw microscopy images of transgenic PF3D7_0702500-3xHA parasites probed with α-HA and α-KAHRP.

**Figure supplement 7.** HSP70x conditional knockout parasites have normal growth at 37 °C and elevated temperatures.

**Figure supplement 7—source data 1.** Raw data file of the parasitaemia fold change of HSP70x knockout infected RBCs at 72 hpi following heat stress or when maintained at 37 °C.

**Figure supplement 7—source data 2.** Raw data file of the parasitaemia fold change of HSP70x knockout infected RBCs at 120 hpi following heat stress or when maintained at 37 °C.

**Figure supplement 8.** The exported protein HSP70x is not required for the trafficking of PfEMP1 onto the infected cell's surface under normal or febrile temperatures.

**Figure supplement 8—source data 1.** Raw data file of the percentage of HSP70x knockout iRBCs positive for PfEMP1 at 24 hpi following heat stress or when maintained at 37 °C.

**Figure supplement 8—source data 2.** Raw data file of the MFI corresponding to surface PfEMP1 levels of HSP70x knockout iRBCs positive for PfEMP1 at 24 hpi following heat stress or when maintained at 37 °C.

**Figure supplement 8—source data 3.** Raw data file of the percentage of HSP70x knockout iRBCs positive for PfEMP1 at 40 hpi following heat stress or when maintained at 37 °C.

**Figure supplement 8—source data 4.** Raw data file of the MFI corresponding to surface PfEMP1 levels of HSP70x knockout iRBCs positive for PfEMP1 at 40 hpi following heat stress or when maintained at 37 °C.

**Figure supplement 9.** HSP70x is not required for functional PSAC activity at 24 hpi under normal or elevated temperatures.

**Figure supplement 9—source data 1.** Raw data file of the percentage of HSP70x knockout iRBCs which are sorbitol sensitive following heat stress or when maintained at 37 °C.

**Figure supplement 10.** PF3D7_0702500 conditional knockout parasites have normal growth at 37 °C and elevated temperatures.

**Figure supplement 10—source data 1.** Raw data file of the parasitaemia fold change of PF3D7_0702500 knockout infected RBCs at 72 hpi following heat stress or when maintained at 37 °C.

**Figure supplement 10—source data 2.** Raw data file of the parasitaemia fold change of PF3D7_0702500 knockout infected RBCs at 120 hpi following heat stress or when maintained at 37 °C.

**Figure supplement 11.** The exported protein PF3D7_0702500 is not required for the trafficking of PfEMP1 onto the infected cell's surface under normal or febrile temperatures.

*Figure 3 continued on next page*

*Figure 3 continued*

**Figure supplement 11—source data 1.** Raw data file of the percentage of PF3D7_0702500 knockout iRBCs positive for PfEMP1 at 24 hpi following heat stress or when maintained at 37 °C.

**Figure supplement 11—source data 2.** Raw data file of the MFI corresponding to surface PfEMP1 levels of PF3D7_0702500 knockout iRBCs positive for PfEMP1 at 24 hpi following heat stress or when maintained at 37 °C.

**Figure supplement 11—source data 3.** Raw data file of the percentage of PF3D7_0702500 knockout iRBCs positive for PfEMP1 at 40 hpi following heat stress or when maintained at 37 °C.

**Figure supplement 11—source data 4.** Raw data file of the MFI corresponding to surface PfEMP1 levels of PF3D7_0702500 knockout iRBCs positive for PfEMP1 at 40 hpi following heat stress or when maintained at 37 °C.

**Figure supplement 12.** PF3D7_0702500 is not required for functional PSAC activity at 24 hours post-invasion (hpi) under normal or elevated temperatures.

**Figure supplement 12—source data 1.** Raw data file of the percentage of PF3D7_0702500 knockout iRBCs which are sorbitol sensitive following heat stress or when maintained at 37 °C.

To test whether disruption of proteins targeted by FIKK kinases during heat stress might help identify required mediators of early trafficking, two additional DiCre-mediated conditional knockout strains of HA-tagged lines were generated. HSP70x (PF3D7_0831700), previously implicated to have a role in PfEMP1 trafficking and heat stress survival, showed increased phosphorylation under heat stress conditions (*Charnaud et al., 2017*; *Day et al., 2019*). PF3D7_0702500, a Maurer's cleft-residing protein (*Jonsdottir et al., 2021*; *Heiber et al., 2013*), exhibited the highest number of phosphosites with increased phosphorylation in response to heat stress. Both tagged proteins localised as previously reported and were efficiently excised upon RAP treatment (*Figure 3—figure supplement 3*; *Figure 3—figure supplement 4*; *Figure 3—figure supplement 5*; *Figure 3—figure supplement 6*; *Charnaud et al., 2017*; *Cobb et al., 2017*; *Jonsdottir et al., 2021*). However, neither HSP70x nor PF3D7_0702500 disruption was found to have an effect on heat stress survival, PfEMP1 trafficking, PSAC surface expression at 37 °C, or was required for accelerated heat stress trafficking (*Figure 3—figure supplement 7*; *Figure 3—figure supplement 8*; *Figure 3—figure supplement 9*; *Figure 3—figure supplement 10*; *Figure 3—figure supplement 11*; *Figure 3—figure supplement 12*).

Collectively, these data show that elevated temperature leads to increased phosphorylation of exported proteins, with the majority of this modification mediated by the Maurer's cleft-localising kinase FIKK10.2. However, neither FIKK4.1, FIKK10.2 nor the putative substrates HSP70x and PF3D7_0702500 are required for the enhanced trafficking of PfEMP1 or the increased sorbitol sensitivity, which is likely driven by elevated PSAC trafficking. Based on these findings, we reasoned that the observed increase in phosphorylation may result from greater protein flux into or through subcellular compartments where FIKK4.1 and FIKK10.2 localise.

## The FIKK10.2-TurboID Maurer's cleft proxiome during normal and febrile temperatures

FIKK10.2 is localised to the Maurer's clefts, which are key trafficking hubs established by the parasite in the RBC. To measure trafficking of proteins through the Maurer's Clefts, we generated a C-terminal fusion of endogenous FIKK10.2 and TurboID (FIKK10.2-TurboID; *Figure 4—figure supplement 1*). In the presence of biotin and ATP, TurboID generates and releases reactive biotin molecules that bind to lysine residues in proximal (up to ~10 nm) proteins (*Branon et al., 2018*). Biotinylated proteins are then digested, and the resulting biotinylated peptides are enriched by immunoprecipitation, quantified by LC-MS/MS and mapped to the corresponding proteins (*Davies et al., 2023*).

To assess how the FIKK10.2-TurboID proxiome changes during heat stress, we measured three conditions: (A) continuous biotin labelling without heat stress to identify proteins that are either persistently or transiently proximal to FIKK10.2 throughout the asexual cycle, (B) biotin labelling from 16 to 24 hpi without heat stress to capture interactions during PfEMP1 trafficking, and (C) biotin labelling with heat stress from 16 to 24 hpi to determine heat-induced changes in FIKK10.2's local protein environment (*Figure 4A–B*).

Across all conditions, a total of 316 unique biotinylated peptides were detected, comprising 278 from *P. falciparum* and 38 from human proteins. In condition A, where biotin was present throughout the parasite life cycle, 256 *P. falciparum* peptides were consistently detected in every replicate. Of

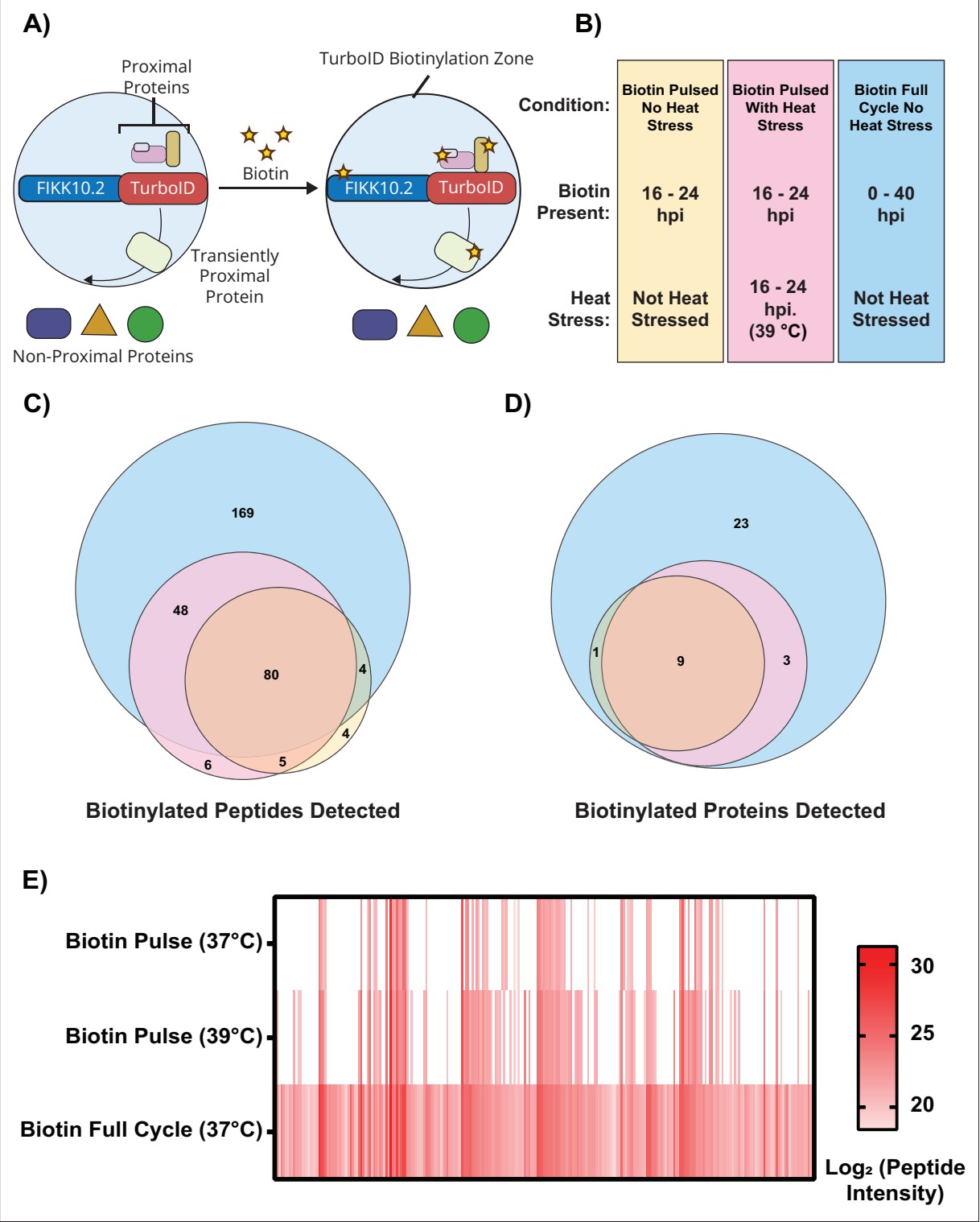

**Figure 4.** FIKK10.2-TurboID reveals changes to the Maurer's cleft protein environment during heat stress. (**A**) In the presence of biotin, transgenic parasites expressing FIKK10.2, a Maurer's cleft residing protein, fused to TurboID were used to biotinylate proximal proteins. (**B**) FIKK10.2-TurboID parasites were cultured under three different conditions: biotin was either present throughout the life cycle, or added as an 8 hr pulse (16–24 hours post-invasion [hpi]) in the presence or absence of heat stress at 39 °C. At 40 hpi, late-stage iRBCs were enriched by Percoll density separation and lysed. Proteins were subsequently digested, and biotinylated peptides were enriched and identified. (**C**) Proportional Venn diagram of detected peptides across the three conditions. (**D**) Proportional Venn diagram of detected proteins in each condition. (**E**) Heat map of biotinylated peptide intensity across

*Figure 4 continued on next page*

*Figure 4 continued*

the three tested conditions. N=3 biological replicates. For (**C**) and (**D**), valid peptides were defined as those detected in at least two replicates, and valid proteins were defined as those with at least two different valid peptides detected.

The online version of this article includes the following source data and figure supplement(s) for figure 4:

**Source data 1.** Raw data file of peptides detected following biotin enrichment of FIKK10.2-TurboID lysates after culturing of the FIKK10.2-TurboID strain in three different conditions.

**Figure supplement 1.** Generation and validation of FIKK10.2-TurboID *P. falciparum* transgenic strain.

**Figure supplement 1—source data 1.** Raw images of the DNA gel assessing the correct 5′ integration of the FIKK10.2-TurboID transgenic parasite strain.

**Figure supplement 1—source data 2.** Raw images of the DNA gel assessing the correct 3′ integration of the FIKK10.2-TurboID transgenic parasite strain.

**Figure supplement 1—source data 3.** Raw images of the DNA gel assessing the absence of the parental strain in the culture of the FIKK10.2-TurboID transgenic parasite strain.

**Figure supplement 1—source data 4.** Raw images of the α-biotin immunoblot of FIKK10.2-TurboID lysates.

**Figure supplement 1—source data 5.** Raw images of the total protein content control of the α-biotin immunoblot of FIKK10.2-TurboID lysates.

**Figure supplement 1—source data 6.** PDF containing the raw images of the DNA gel assessing the correct 5′ integration of the FIKK10.2-TurboID transgenic parasite strain.

**Figure supplement 1—source data 7.** PDF containing the raw images of the DNA gel assessing the correct 3′ integration of the FIKK10.2-TurboID transgenic parasite strain.

**Figure supplement 1—source data 8.** PDF containing the raw images of the DNA gel assessing the absence of the parental strain in the culture of the FIKK10.2-TurboID transgenic parasite strain.

**Figure supplement 1—source data 9.** PDF containing the raw images of the α-biotin immunoblot of FIKK10.2-TurboID lysates.

**Figure supplement 1—source data 10.** PDF containing the raw images of the total protein content control of the α-biotin immunoblot of FIKK10.2-TurboID lysates.

**Figure supplement 2.** Predicted Maurer's cleft protein topology from enriched phosphopeptides and biotinylated peptides.

these, 231 peptides (90%) mapped to 39 exported proteins, underscoring the specificity and robustness of this peptide-level enrichment approach (*Davies et al., 2023*). These proteins are very likely to be either trafficked through or localised to the Maurer's clefts (*Supplementary files 4 and 5*). Many of these proteins have predicted transmembrane domains, and by mapping the FIKK10.2-TurboID enriched peptides on their projected topology, the RBC cytosol-facing domain can be predicted (*Figure 4—figure supplement 2*).

Although six peptides were detected only in the 39 °C heat-stressed biotin-pulsed condition (*Figure 4C*), they originate from proteins also present in the other conditions (*Figure 4D–E*). Thus, these TurboID data indicate that FIKK10.2 does not encounter unique proteins during heat stress that are absent under normal conditions. Based on these data, we hypothesised that a greater number of exported proteins are in proximity to FIKK10.2 during febrile heat stress, suggesting increased trafficking of proteins to or through the Maurer's clefts. These proteins are phosphorylated by FIKK10.2 during their transit. One possible origin of increased trafficking of exported proteins could be increased translocation of proteins through the PVM into the RBC cytosol.

## Febrile heat stress increases PfEMP1 (VAR2CSA) trafficking into the RBC cytosol and enhances its surface presentation on a greater proportion of iRBCs

To directly assess whether PfEMP1 is trafficked earlier to the RBC surface during heat stress, we endogenously tagged the VAR2CSA variant with NanoLuc (*Figure 5A*; *Figure 5—figure supplement 1*; *England et al., 2016*; *Looker et al., 2022*). Following heat stress, a significantly greater proportion of iRBCs displayed surface-localised PfEMP1 tagged with NanoLuc, as detected by α-VAR2CSA staining and flow cytometry, indicating that the fusion protein retains its capacity to traffic into the host cell and reach the iRBC surface (*Figure 5B*).

To quantify export efficiency under normal (37 °C) and heat-stressed (39 °C) conditions, we measured NanoLuc activity across three distinct compartments: (A) the RBC cytosol, (B) the parasitophorous vacuole (PV), and (C) the total parasite/RBC lysate. These compartments were selectively isolated in

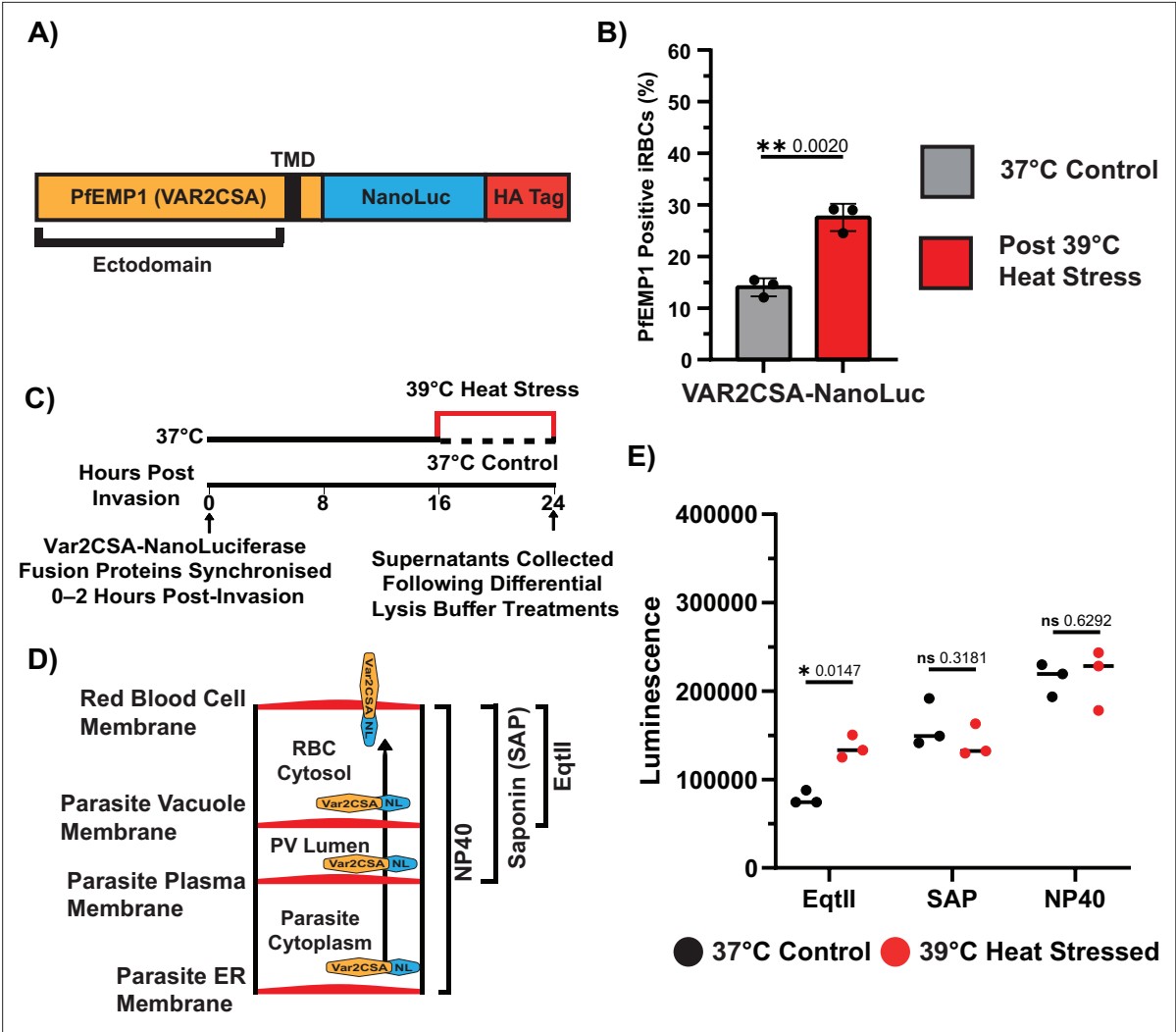

**Figure 5.** Febrile heat stress enhances trafficking of endogenously tagged PfEMP1 (VAR2CSA) to the RBC cytosol and surface. (**A**) A transgenic *P. falciparum* strain expressing NanoLuc fused to the C-terminus of VAR2CSA was generated. (**B**) Following heat stress, a greater proportion of VAR2CSA-NanoLuc expressing parasites displayed PfEMP1 on the iRBC surface compared to parasites maintained at 37 °C. N=3 biological replicates. Error bars represent ±1 SD. Statistical significance was determined using an unpaired t-test on log-transformed data with Welch's correction. (**C**) At 24 hpi, following heat stress or continuous culture at 37 °C, cells were differentially permeabilised using three treatments. NanoLuc activity was measured in the supernatants. (**D**) Overview of the compartments released by each permeabilisation treatment: EqtII permeabilises the RBC membrane while preserving the PVM, SAP releases proteins from the RBC cytosol and PV lumen, and NP40 solubilises all non-nuclear proteins from the parasite, PV, and RBC cytosol. (**E**) Significantly higher NanoLuc activity was detected in the supernatant of heat-stressed parasites only following EqtII permeabilisation. Assays were performed at room temperature. Data represent the mean of three biological replicates, each averaged from three technical replicates. Statistical significance was determined using the one-way ANOVA test with the Benjamini, Krieger, and Yekutieli FDR correction.

The online version of this article includes the following source data and figure supplement(s) for figure 5:

**Source data 1.** Raw data file of the percentage of VAR2CSA-NanoLuc expressing iRBCs which are positive for cell surface PfEMP1 (VAR2CSA) following heat stress compared to iRBCs maintained at 37 °C.

**Source data 2.** Raw data file of the NanoLuc luminescence detected in different lysis buffer treatments of VAR2CSA-NanoLuc expressing iRBCs following heat stress compared to iRBCs maintained at 37 °C.

**Figure supplement 1.** Generation and validation of VAR2CSA-NanoLuc-3xHA *P. falciparum* transgenic strain.

**Figure supplement 1—source data 1.** Raw images of the DNA gel assessing the correct 5' integration of the VAR2CSA-NanoLuc-3xHA transgenic parasite strain.

**Figure supplement 1—source data 2.** Raw images of the DNA gel assessing the correct 3' integration of the VAR2CSA-NanoLuc-3xHA transgenic parasite strain.

**Figure supplement 1—source data 3.** Raw images of the DNA gel assessing the absence of the parental strain in the culture of the VAR2CSA-

*Figure 5 continued on next page*

*Figure 5 continued*

NanoLuc-3xHA transgenic parasite strain.

**Figure supplement 1—source data 4.** Raw images of the α-HA immunoblot of VAR2CSA-NanoLuc-3xHA lysates.

**Figure supplement 1—source data 5.** Raw images of the total protein content control of the α-HA immunoblot of VAR2CSA-NanoLuc-3xHA lysates.

**Figure supplement 1—source data 6.** PDF containing the raw images of the DNA gel assessing the correct 5′ integration of the VAR2CSA-NanoLuc-3xHA transgenic parasite strain.

**Figure supplement 1—source data 7.** PDF containing the raw images of the DNA gel assessing the correct 3′ integration of the VAR2CSA-NanoLuc-3xHA transgenic parasite strain.

**Figure supplement 1—source data 8.** PDF containing the raw images of the DNA gel assessing the absence of the parental strain in the culture of the VAR2CSA-NanoLuc-3xHA transgenic parasite strain.

**Figure supplement 1—source data 9.** PDF containing the raw images of the α-HA immunoblot of VAR2CSA-NanoLuc-3xHA lysates.

**Figure supplement 1—source data 10.** PDF containing the raw images of the total protein content control of the α-HA immunoblot of VAR2CSA-NanoLuc-3xHA lysates.

the supernatant following different permeabilisation protocols. Equinatoxin II (EqtII) permeabilises the RBC membrane while leaving the PVM intact, saponin (SAP) releases proteins from both the RBC cytosol and PV lumen (*Bullen et al., 2022*), and NP40 detergent solubilises all non-nuclear proteins within the parasite, PV, and RBC cytosol (*Figure 5C–D*).

Importantly, NanoLuc activity in the NP40 supernatant did not differ significantly between control and heat-stressed conditions, suggesting that total VAR2CSA expression remains unchanged (*Figure 5E*). This supports our hypothesis that *P. falciparum* does not markedly accelerate its developmental program or significantly alter protein expression in response to heat stress, and that increased cytoadhesion under febrile conditions is more likely due to altered protein trafficking.

Notably, following heat stress, NanoLuc activity significantly increased (58% more) in the EqtII permeabilised supernatant but not in the SAP fraction (*Figure 5E*). This suggests that heat stress promotes trafficking of VAR2CSA from the PV into downstream compartments such as the Maurer's clefts and ultimately to the RBC surface.

## Transmembrane domain-containing exported reporter proteins are trafficked more efficiently into the RBC cytosol during febrile heat stress

To evaluate whether the enhanced trafficking of exported proteins under heat stress (39 °C) is a general phenotype or restricted to transmembrane domain-containing proteins such as PfEMP1 and the PSAC component CLAG, we generated four NanoLuc reporter constructs with variable N-terminal protein fusions (*Figure 6—figure supplement 1*). All four reporters were expressed under the constitutive HSP90 (PF3D7_0708400) promoter and integrated into the EBA165 pseudogene locus (PF3D7_0424300).

To first test whether other TMD-containing proteins are differentially trafficked into the RBC during febrile heat stress, the export of PF3D7_0702500-NanoLuc was measured. PF3D7_0702500 is a Maurer's cleft-resident protein (*Jonsdottir et al., 2021*; *Heiber et al., 2013*), which we found to be significantly more phosphorylated and more biotinylated by FIKK10.2-TurboID following heat stress (*Supplementary file 1 and 4*). The trafficking of PF3D7_0702500-NanoLuc into the iRBC cytosol mirrored that observed for VAR2CSA-NanoLuc, with PF3D7_0702500-NanoLuc being moderately more abundant (30%) in the EqtII fraction following heat stress (39 °C) compared to normal conditions (37 °C; *Figure 6A–C*).

As a non-TMD-containing protein reporter, we generated REX3-NanoLuc, which contains the first 61 amino acids of REX3 that are sufficient for export (*Sargeant et al., 2006*; *Tarr et al., 2014*). This reporter did not show a significant increase in abundance in any of the three permeabilisation fractions following heat stress (*Figure 6D*).

These data suggest that TMD-containing proteins are specifically affected in trafficking under heat conditions. To directly test this, the TMD from PF3D7_0702500 was added to REX3-NanoLuc to generate REX3-TMD-NanoLuc. Trafficking of REX3-TMD-NanoLuc into the RBC cytosol was less efficient than REX3-NanoLuc at 37 °C. However, following heat stress, 57.6% more REX3-TMD-NanoLuc was detected in the RBC cytosolic fraction compared to normal temperature conditions (*Figure 6E*). A

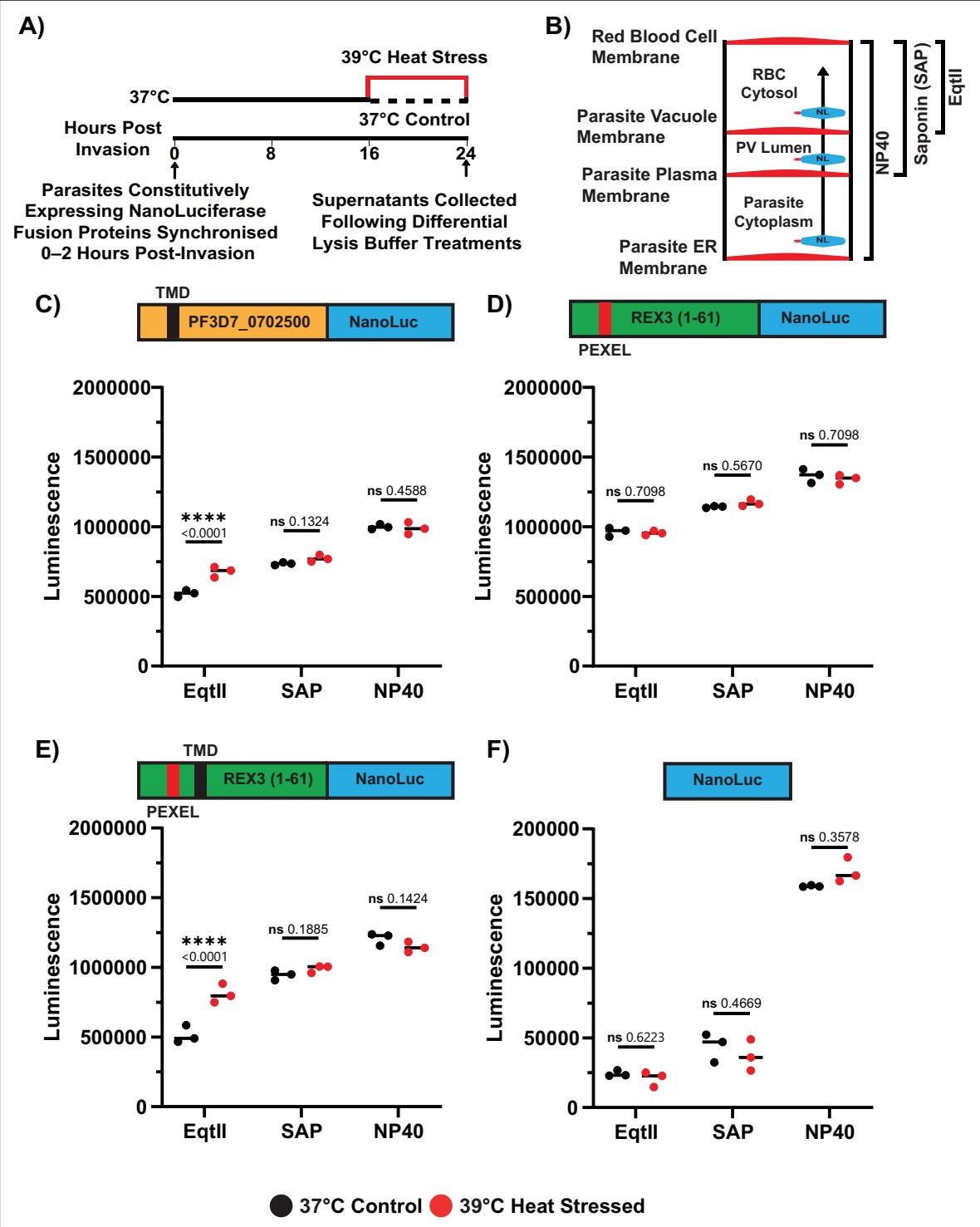

**Figure 6.** Constitutively expressed exported NanoLuc strains identify transmembrane domains as key determinants for increased export into the host cell during febrile temperatures. (**A**) Tightly synchronised *P. falciparum* parasites constitutively expressing one of four NanoLuc reporter constructs were exposed to heat stress (39 °C) or maintained at 37 °C between 16 and 24 hpi. At 24 hpi, cells were subjected to three different differential permeabilisation treatments. NanoLuc activity in the resulting supernatants was measured. (**B**) Schematic representation of the cellular compartments released by each permeabilisation method. All assays were performed at room temperature. (**C**) The constitutively expressed PF3D7_0702500-NanoLuc reporter, but not (**D**) the REX3-NanoLuc reporter, showed significantly increased NanoLuc activity in the EqtII-permeabilised supernatant following heat stress. REX3-NanoLuc contains the first 61 amino acids of REX3, which are sufficient to mediate export (*Sargeant et al., 2006*; *Tarr et al., 2014*). (**E**) Fusion of the PF3D7_0702500 TMD to the REX3-NanoLuc reporter reduced luminescence in the EqtII-permeabilised supernatant under non-stress

*Figure 6 continued on next page*

*Figure 6 continued*

conditions. However, increased luminescence was observed following heat stress. (**F**) Reporters lacking an N-terminal fusion to an exported *P. falciparum* protein showed negligible luminescence in the supernatant following SAP or EqtII treatment. Across all four constructs, no significant differences in luminescence were observed between SAP and NP40-permeabilised samples when comparing conditions with and without heat stress. Data represent the mean of three biological replicates, each averaged from three technical replicates. Statistical significance was determined using the one-way ANOVA test with the Benjamini, Krieger, and Yekutieli FDR correction.

The online version of this article includes the following source data and figure supplement(s) for figure 6:

**Source data 1.** Raw data file of the NanoLuc luminescence detected in different lysis buffer treatments of PF3D7_0702500-NanoLuc expressing iRBCs following heat stress compared to iRBCs maintained at 37 °C.

**Source data 2.** Raw data file of the NanoLuc luminescence detected in different lysis buffer treatments of REX3-NanoLuc expressing iRBCs following heat stress compared to iRBCs maintained at 37 °C.

**Source data 3.** Raw data file of the NanoLuc luminescence detected in different lysis buffer treatments of REX3-(TMD)-NanoLuc expressing iRBCs following heat stress compared to iRBCs maintained at 37 °C.

**Source data 4.** Raw data file of the NanoLuc luminescence detected in different lysis buffer treatments of the NanoLuc only control expressing iRBCs following heat stress compared to iRBCs maintained at 37 °C.

**Figure supplement 1.** Generation and validation of constitutively expressed NanoLuc reporter strains.

**Figure supplement 1—source data 1.** Raw images of the DNA gel assessing the correct 5′ integration of the four NanoLuc reporter transgenic parasite strains.

**Figure supplement 1—source data 2.** Raw images of the DNA gel assessing the correct 3′ integration of the four NanoLuc reporter transgenic parasite strains.

**Figure supplement 1—source data 3.** PDF containing the raw images of the DNA gel assessing the correct 5′ integration of the four NanoLuc reporter transgenic parasite strain.

**Figure supplement 1—source data 4.** PDF containing the raw images of the DNA gel assessing the correct 3′ integration of the four NanoLuc reporter transgenic parasite strain.

non-exported NanoLuc strain was also generated as a negative control and showed negligible signal in the exported compartments, with no significant change under heat stress (*Figure 6F*).

Collectively, these data suggest that elevated febrile temperatures alleviate a trafficking bottleneck between the PV and the RBC cytosol, at least for the TMD-containing proteins examined in this study. Notably, the addition of the PF3D7_0702500's TMD to a soluble reporter was sufficient to enhance export under heat stress, indicating that this may represent a broader mechanism by which TMDs facilitate temperature-responsive trafficking into the RBC cytosol.

## Discussion

The mechanism underlying increased iRBC cytoadhesion following febrile heat stress is not well understood, with both PfEMP1-dependent and independent pathways proposed. Variability in experimental heat stress conditions, including differences in temperature that affect parasite viability, as well as differences in duration and timing within the asexual cycle, likely contributes to the divergent findings reported across studies. We show that elevated temperatures at multiple stages of the parasite life cycle negatively affect parasite viability. These growth-impairing conditions are likely to trigger significant stress or apoptosis-like responses. To minimise the potential for such stress responses to confound cytoadhesion outcomes, we selected a non-destructive heat stress condition (39 °C, 16–24 hpi) for our experiments. More extreme protocols may not accurately reflect the febrile temperatures typically experienced by malaria patients. In addition to the PfEMP1-dependent mechanism examined here, increased phosphatidylserine (PS) on the surface of iRBCs following 1 hr at 40 °C has been observed and coincides with increased cytoadhesion to CHO cells expressing endothelial receptors (*Zhang et al., 2018*). Although a role for PfEMP1 in this increased cytoadhesion was not identified previously (*Zhang et al., 2018*), our findings indicate that PfEMP1 surface expression contributes possibly alongside PS to the heightened cytoadhesion during fever. However, genetic deletion of proteins required for PS exposure and PfEMP1 surface translocation may be necessary to unequivocally establish the individual contribution of each to overall cytoadhesion.

Our data confirm a previous study reporting increased PfEMP1 on the RBC surface following febrile heat stress (*Udomsangpetch et al., 2002*). We propose that this increase in surface trafficking results

from a reduction in a trafficking bottleneck during elevated temperature. Elevated temperature does not only affect PfEMP1 surface translocation but also enhances trafficking of other TMD-containing proteins into the RBC. This likely results in increased PSAC levels at the iRBC surface, although we cannot fully exclude the possibility that the observed increase in sorbitol sensitivity is due to enhanced PSAC activity following heat stress, for example through post-translational modifications. However, we consider this explanation less likely, as deletion of the kinases known to phosphorylate PSAC components had no measurable effect on sorbitol sensitivity. Using several complementary approaches, including NanoLuc reporter assays, proteomics, DNA content measurements, egress timing, and assessments of subsequent replication, we conclude that under our heat stress conditions (39 °C, 16–24 hpi), parasite development does not accelerate. These findings indicate that the observed phenotypes are due to differences in protein trafficking.

While heat stress induces changes in the Maurer's cleft proxiome and phosphoproteome, we did not identify any proteins that were altered exclusively under heat stress compared to 37 °C. Later in the asexual cycle, heat-stressed iRBCs reach levels of PfEMP1 surface expression comparable to those maintained at 37 °C (*Figure 3—figure supplement 8*; *Figure 3—figure supplement 11*). This suggests that proteins involved in accelerated trafficking during heat stress are also present and likely functional at normal temperatures, rather than being uniquely expressed or specifically localised in response to elevated temperature. However, it is worthwhile to note that some phosphorylation sites increase upon heat stress that do not depend on FIKK10.2 nor FIKK4.1. Several of these sites could also not be attributed to previously identified FIKK substrates, indicating that some may be mediated by FIKKs other than FIKK4.1 and FIKK10.2, or by host cell kinases.

The mechanism by which proteins are differentially exported at elevated temperatures remains unknown, although differential trafficking requirements for TMD and non-TMD-containing proteins have been proposed at both the PPM and PVM (*Matthews et al., 2019*; *Beck and Ho, 2021*). Our reporter assay measures export into the RBC compartment but cannot definitively distinguish whether the trafficking differences occur at the level of the PVM or later, for example between the Maurer's clefts and the RBC plasma membrane. Nevertheless, the PTEX complex, which mediates protein translocation from the PV through the PVM into the RBC cytosol (*de Koning-Ward et al., 2009*), is a likely candidate to facilitate increased export of TMD-containing proteins under heat stress. We excluded HSP70x from being involved in protein trafficking, as we show that conditional deletion of HSP70x does not affect PfEMP1 surface translocation.

What may be the consequences of accelerated PfEMP1 trafficking, or protein surface export during fever in a human infection? Our data show that a greater proportion of iRBCs are sensitive to sorbitol following heat stress, suggesting that more cells may present functional PSAC at the surface. Increased function of PSAC could potentially enhance nutrient uptake under these strained conditions. Elevated PfEMP1 levels at the iRBC surface during heat stress may help counteract the negative effects of increased cellular rigidity that occur at higher temperatures (*Zhang et al., 2018*). As cytoadhesion is thought to prevent filtration of the rigidified iRBC in the spleen (*David et al., 1983*), concurrent increased cytoadhesion may need to occur with increased stiffness. How altered protein trafficking during heat stress affects the interplay between iRBC stiffness and their potential to sequester and obstruct small vessels remains to be addressed.

If increased body temperature leads to an increased cytoadhering parasite load, malaria symptoms may be exacerbated. The observed reduction in the age of circulating parasites at higher temperatures suggests that cytoadhesion may occur earlier under febrile conditions. An alternative explanation is that the younger parasite age reflects a recent, synchronised egress event, which has been proposed to trigger fever (*Oakley et al., 2011*). However, tightly synchronised parasite replication is likely rare in fever-inducing natural infections.

Quantifying the synchronicity of natural *P. falciparum* infections is inherently difficult, primarily due to the sequestration of mature parasite stages via cytoadherence and the ethical concerns of continuous monitoring without treatment. Controlled human malaria infection (CHMI) studies, such as *Kapulu et al., 2021*, sampled circulating parasites twice daily using quantitative PCR. Infections were allowed to progress for up to 21 days or until fever symptoms developed. Even in these tightly controlled conditions, circulating parasites are often detectable at every 12 hr interval, suggesting that they are not tightly synchronised. As parasite burden increases, complete absence of circulating parasites becomes increasingly rare, further indicating asynchronous development. Polyclonal

infections, resulting from repeated mosquito bites and the introduction of genetically distinct parasite strains, also appear to drive asynchronicity (*Ciuffreda et al., 2020*). This is particularly relevant in high-transmission settings where individuals may receive, on average, more than 20 Anopheles mosquito bites per day (*Sangbakembi-Ngounou et al., 2022*). Importantly, *P. falciparum* fever should not be conflated with synchronicity. As noted by *Kitchen, 1949*, the febrile patterns of *P. falciparum* are so diverse that the term "atypical" may not meaningfully be applied. While fever may influence parasite dynamics, it is unlikely that only specific stages experience it; all stages are plausibly exposed to host fever responses during an infection.

Ultimately, it is unclear if febrile conditions promote survival of the parasite or the host. It is likely that a sustained fever (>39 °C, >8 hr) restricts parasite proliferation. However, the frequency of sustained high malarial fevers is not well documented. Moreover, data investigating parasite resilience to febrile heat stress are based mainly on lab-adapted parasites, which are not under selection pressure to survive elevated temperatures. Antipyretic drugs reduce fever, but whether they should be used during viral and bacterial infections is uncertain (*Ludwig and McWhinnie, 2019*). This ambiguity also extends to malaria, where a meta-analysis of several clinical studies investigating malaria patient outcome following antipyretic treatment found that there was insufficient data to confirm or refute an impact of antipyretic measures on parasitaemia or malarial illness (*Meremikwu et al., 2000*). However, a recent clinical trial showed a positive correlation of antipyretic treatment with host survival in malaria infection, indicating further studies in this area are warranted (*Birbeck et al., 2024*). Furthermore, a recent pre-print demonstrates increased binding of iRBCs to blood vessels using 3D models as a result of fever conditions (*Introini et al., 2025*). Together, increased PfEMP1 on the surface earlier with increased host-cell binding would exacerbate the negative effects on fever on host survival if the overall impact of fever on parasite survival is not taken into account. To directly test the impact of fever on parasite survival versus cytoadhesion, well-controlled studies in human populations will be required in the future.

# Materials and methods

**Key resources table**

| Reagent type (species) or resource | Designation | Source or reference | Identifiers | Additional information |
|---|---|---|---|---|
| Strain, strain background (*Plasmodium falciparum*) | NF54::DiCre | Tibúrcio et al., PMID:31530668 | | Parental transgenic line used throughout this manuscript. |
| Strain, strain background (*P. falciparum*) | NF54 | Tibúrcio et al., PMID:31530668 | | Isolate predicted to originate from West Africa. |
| Strain, strain background (*P. falciparum*) | HB3 | Gift (Maria Bernabeu) | | Honduras isolate |
| Strain, strain background (*P. falciparum*) | Cam3.II | Gift (Colin Sutherland) | | Cambodia isolate |
| Strain, strain background (*P. falciparum*) | HL2208 | Gift (Colin Sutherland) PMID:38019958 | | Uganda isolate |
| Cell line (*Homo sapiens*) | Human erythrocytes | UK, NHS Blood and Transplant (NHSBT) | | RBCs used for *P. falciparum* parasite culture |
| Genetic reagent (*P. falciparum*) | FIKK1 conditional knockout strain | Davies et al., PMID:32284562 | | HA-tagged DiCre excisable line |
| Genetic reagent (*P. falciparum*) | FIKK4.1 conditional knockout strain | Davies et al., PMID:32284562 | | HA-tagged DiCre excisable line |
| Genetic reagent (*P. falciparum*) | FIKK4.2 conditional knockout strain | Davies et al., PMID:32284562 | | HA-tagged DiCre excisable line |
| Genetic reagent (*P. falciparum*) | FIKK9 family conditional knockout strain | Davies et al., PMID:32284562 | | DiCre excisable line to knockout all FIKK9 genes |
| Genetic reagent (*P. falciparum*) | FIKK10.1 conditional knockout strain | Davies et al., PMID:32284562 | | HA-tagged DiCre excisable line |

*Continued on next page*

*Continued*

| Reagent type (species) or resource | Designation | Source or reference | Identifiers | Additional information |
|---|---|---|---|---|
| Genetic reagent (*P. falciparum*) | FIKK10.2 conditional knockout strain | Davies et al., PMID:32284562 | | HA-tagged DiCre excisable line |
| Genetic reagent (*P. falciparum*) | FIKK10.2-TurboID-V5 strain | This paper | | Endogenous FIKK10.2 C-terminally fused to TurboID-V5 (NF54::DiCre is the parental strain) |
| Genetic reagent (*P. falciparum*) | HSP70x conditional knockout strain | This paper | | HA-tagged DiCre excisable line (NF54::DiCre is the parental strain) |
| Genetic reagent (*P. falciparum*) | PF3D7_0702500 conditional knockout strain | This paper | | HA-tagged DiCre excisable line (NF54::DiCre is the parental strain) |
| Antibody | Anti-VAR2CSA (human monoclonal, PAM7.5) | Gift (Lars Hviid) | | 1:200 (flow cytometry) |
| Antibody | Goat anti-human PE-conjugated secondary | Thermo Fisher Scientific | Cat#: 12-4998-82, RRID:AB_465926 | 1:200 (flow cytometry) |
| Antibody | Anti-HA (rat monoclonal, clone 3F10) | Merck | Cat#: 11867423001, RRID:AB_390918 | 1:3000 (western blot) |
| Antibody | Streptavidin-HRP | Pierce | Cat#: 21130 | 1:5000 (western blot) |
| Antibody | Anti-SBP1 | Gift (Tobias Spielmann) | | Parasite protein detection, 1:5000 (Immunofluorescence Microscopy) |
| Antibody | Anti-KAHRP | European Malaria Reagent Repository | | Parasite protein detection, 1:2000 (Immunofluorescence Microscopy) |
| Chemical compound, drug | WR99210 | Jacobus Pharmaceuticals | | Parasite drug selection |
| Chemical compound, drug | G418 | Gibco | Cat#: 10131027 | Parasite drug selection |
| Chemical compound, drug | Rapamycin (RAP) | Sigma | Cat#: 553210–5 MG | 100 nM (DiCre induction to induce conditional knockout) |
| Chemical compound, drug | Furosemide | Sigma | Cat#: F4381-10G | PSAC inhibitor |

## In vitro maintenance and synchronisation of parasites

Human erythrocytes infected with asexual stages of *P. falciparum* were cultured at 37 °C in complete medium. Complete medium was prepared using 1 L RPMI-1640, supplemented with 5 g AlbuMAX II (Thermo Fisher Scientific), 0.292 g L-glutamine, 0.05 g hypoxanthine, 2.3 g sodium bicarbonate, 0.025 g gentamicin and 5.957 g HEPES. Washed and purified RBCs were provided by the NHS Blood and Transplant (NHSBT) services. Parasites were cultured following standard procedures in a gas atmosphere of 90% N2, 5% $CO_2$, and 5% $O_2$ (*Trager and Jensen, 1976*). Parasite cultures were synchronised using Percoll (GE Healthcare) to isolate mature schizont-stage parasites. The purified schizonts were incubated at 37 °C in complete medium with fresh RBCs for 2–4 hr in a shaking incubator. A second Percoll purification was performed to remove any remaining schizonts, and the remaining cells were resuspended in 5% sorbitol for 10 min at 37 °C before being washed into complete medium. These two steps resulted in highly synchronous parasites with minimal debris in the solution.

## Generation of transgenic *P. falciparum* strains

*P. falciparum* transfections were performed as described before (*Collins et al., 2013*; *Jones et al., 2016*). Transgenic *Plasmodium falciparum* NF54::DiCre (*Tibúrcio et al., 2019*) parasite lines were generated using CRISPR-Cas9. Unless otherwise specified, the NF54::DiCre parental strain was used as a wild type-like reference throughout this manuscript. Suitable guide RNAs (gRNAs) were identified using the Eukaryotic Pathogen CRISPR guide RNA/DNA Design Tool (*Peng and Tarleton, 2015*). Complementary oligonucleotides corresponding to the 19 nucleotides nearest the selected PAM sequence were synthesised (Integrated DNA Technologies, IDT), phosphorylated using T4 polynucleotide kinase, annealed and ligated into the pDC_Cas9_hDHFRyFCU (*Knuepfer et al., 2017*) vector digested with BbsI. To produce compatible sticky ends between the annealed oligonucleotides and

the BbsI-digested vector, the forward oligonucleotide included a 5′-ATTG overhang, and the reverse oligonucleotide included a 5′-AAAC overhang. gRNAs targeting the gene of interest were assembled using appropriate oligonucleotide pairs. Repair templates were designed in three different configurations. For conditional knockouts, templates included LoxP-containing introns flanking a recodonised, HA-tagged gene followed by a T2A skip peptide and a drug selection cassette (*Jones et al., 2016*). For TurboID fusions, templates encoded a recodonised gene C-terminally fused to TurboID with a V5 tag, followed by a T2A skip peptide and drug selection marker. For NanoLuc reporters, the construct comprised the HSP90 promoter driving expression of a variable N-terminal domain fused to NanoLuciferase, followed by a T2A skip peptide and a selection cassette (*Jones et al., 2016*). Gene blocks encoding recodonised genes, TurboID, and NanoLuciferase were synthesised (IDT). All repair templates were flanked by gene-specific homology arms and included restriction sites to enable excision of the linear repair templates. Prior to transfection, 20 μg of a Cas9-expressing guide plasmid and 20 μg of a linearised repair template plasmid were ethanol precipitated. Ethanol precipitation was performed by adding 0.1 volume of 3 M Sodium Acetate (pH 5.2) and 2 volumes of 100% ethanol, followed by at least 2 hr incubation at –20 °C. DNA was subsequently centrifuged at 4 °C for 30 min and washed twice with 70% ethanol. DNA pellets were air-dried and resuspended in 10 μL Tris-EDTA (TE) buffer. Synchronised egressing schizonts (NF54 DiCre) were enriched by 60% Percoll before washing in complete medium. The resuspended DNA was mixed with 90 μL P3 Primary cell solution (Lonza) and was used to resuspend egressing schizonts before transfer to transfection cuvette.

Transfection was performed by electroporation using the FP158 programme of an Amaxa 4D Electroporator machine (Lonza). Following transfection, the parasites were transferred to pre-warmed flasks containing 2 mL complete medium and 300 μL fresh uRBCs. After 40 min of gentle shaking in an incubator at 37 °C, 8 mL complete medium was added to each flask. Transfected parasites were cultured without selection for 24 hr, then selection was first performed with 2.5 nM WR99210 (Jacobus Pharmaceuticals) for 48 hr. After WR99210-resistant parasites emerged, they were further selected with 225 μg/mL G418 (Gibco). Correct integration of the repair template into the parasite's DNA was confirmed by overlapping PCR of the 5′ and 3′ regions (*Supplementary file 6*).

## Conditional knockout of DiCre-expressing parasites

To induce DiCre-mediated *LoxP* site recombination, synchronised ring-stage parasites were treated with either 100 nM RAP (Sigma) or an equivalent volume of DMSO to act as a control. After 4 hr at 37 °C, iRBCs were washed twice with RPMI and returned to culture in complete medium. Samples for DNA and protein extraction were taken at least one replication cycle (48 hr) after RAP treatment.

## PfEMP1 (VAR2CSA) cell-surface detection

50 μL of iRBC culture was centrifuged (3000 × *g*, 1 min) and pelleted cells were blocked in 200 μL 3% BSA in PBS for 15 min at room temperature. Cultures were then resuspended in 50 μL of 1% BSA in PBS with 1:200 α-VAR2CSA (*Barfod et al., 2007*; PAM7.5) human monoclonal antibody (a kind gift from Lars Hviid) and incubated for 30 min at 37 °C. Cells were then pelleted and washed in 1 mL of PBS three times to remove unbound primary antibody. Cultures were then resuspended in 50 μL of 1% BSA in PBS with 1:200 α-Human PE-conjugated secondary antibody (12-4998-82, Thermo Fisher), 1:2000 Hoechst (Thermo Fisher) and incubated for 30 min at 37 °C. Cells were then pelleted and washed in 1 mL of PBS three times to remove unbound secondary antibody. Labelled cells were then measured by flow cytometry (FACSVerse, BD Biosciences) without fixation.

## Application of heat stress *to P. falciparum* cultures

To heat stress parasites, separate cell culture incubators were set to different temperatures. The correct temperature was determined independently of the incubator screen display by multiple probes. When required, flasks were moved from one incubator to another to heat stress or to return to 37 °C after heat stress. An effort was made to not keep flasks out of the incubator for long, and flask transfer took less than 30 s. When heat stress was being applied, the opening of the incubators was kept to a minimum.

To measure the temperature dynamics of medium within heat-stressed flasks, a temperature logging probe (Fisherbrand 90009–36) was inserted into the media, and the temperature was recorded every 30 s (Accuracy: ±0.5 °C, Resolution: 0.1 °C).

## Infected RBC cytoadhesion assay

Cytoadhesion assays were performed as described previously (*Davies et al., 2023*). 100 µL of CSA (Sigma-Aldrich; 1 mg/mL in PBS) was added to each channel of an untreated Ibidi µ-slide VI$_{0.4}$ and left to incubate overnight at 4 °C. The channels were blocked with 3% BSA in PBS for one hour at room temperature before three washes in PBS and one final wash in RPMI without Albumax. Parasites were synchronised to a two-hour invasion window (10% parasitaemia, 1% haematocrit). At 24 and 40 hpi, 500 µL of iRBC culture was resuspended in 4 mL of RPMI without Albumax and drawn through an Ibidi µ-slide VI$_{0.4}$ previously coated with CSA and blocked in BSA. Using a Harvard syringe pump (Sigma), 1 dyne/cm$^2$ (0.1 Pa) shear stress was applied across each channel (0.55 mL/min). At the same pressure and flow rate, the channels were then washed with 4 mL of RPMI without Albumax containing Hoechst DNA stain (1:2000, Thermo Fisher). Ten frames (40 x) across the centre of the microchannel were taken with Nikon Eclipse Ti2 Microscope. Cytoadherent cells were quantified via ImageJ imaging software as Hoechst-positive cells. Multiple conditions were measured in parallel, and binding was normalised to the 37 °C control.

## *P. falciparum* growth assays

To assess parasite growth, cultures were synchronised to a 2 hr invasion window and parasitaemia was adjusted to 0.5–1%. Parasitaemia was typically measured in the initial cycle (cycle 1) and the subsequent two cycles using flow cytometry. At 24 (cycle 1), 72 (cycle 2), and 120 (cycle 3) hpi, a 50 µL aliquot of iRBC culture (1–4% haematocrit) was mixed thoroughly with 50 µL of 2×glutaraldehyde (GA; 0.4%) in a 96-well plate. Plates were sealed and stored at 4 °C. Once all time points were collected, 50 µL was removed from each fixed well and combined with 50 µL of 2×SYBR Green DNA stain (1:5000, Thermo Fisher) in PBS, then incubated at 37 °C for 30 min. Following incubation, 100 µL of PBS was added to each well, and the plate was analysed using a high-throughput plate attachment on the FACSVerse flow cytometer (BD Biosciences). Infected cells were gated based on SYBR Green fluorescence intensity, and parasitaemia was calculated for each time point.

## Infected RBC sorbitol sensitivity assay

Parasites were synchronised to a 2 hr invasion window (1–10% parasitaemia and 1–5% haematocrit). To assess sorbitol sensitivity of iRBCs, two 200 µL aliquots of iRBC culture were taken from each sample and centrifuged at 3000 × *g* for 2 min. The cell pellets were washed with 1 mL of PBS and centrifuged again. Each aliquot was then resuspended in either 5% sorbitol or PBS, with both solutions pre-warmed to 37 °C. Following a 10-min incubation at 37 °C, cells were washed four additional times with 1 mL of PBS. Pelleted cells were either fixed and parasitaemia determined by SYBR Green staining (1:10,000, Thermo Fisher), or analysed directly without fixation. Flow cytometry was performed as described previously using a FACSVerse flow cytometer (BD Biosciences). The percentage of sorbitol-sensitive parasites was calculated using the following formula: $\left(1 - \left(\frac{Parasitaemia\ Post\ Sorbitol\ Treatment}{Parasitaemia\ Post\ PBS\ Treatment}\right)\right) * 100$. To assess whether sorbitol-mediated lysis was restricted to iRBCs and inhibited by the PSAC inhibitor furosemide following heat stress, both non-heat-stressed and heat-stressed parasite cultures were treated with 200 µM furosemide for 10 min at 37 °C. Cultures were maintained at 2% haematocrit with parasitaemia greater than 10%. Following pre-treatment with furosemide, cultures were resuspended in 5% sorbitol in PBS with 200 µM furosemide maintained. After a 10-min incubation, samples were centrifuged, and haemoglobin release was quantified by measuring the absorbance of the supernatant at 415 nm using a plate reader. As additional controls, cultures were incubated in PBS alone without sorbitol or treated with saponin to achieve complete cell lysis. Supernatants from saponin-treated samples were diluted fivefold to ensure measurements fell within the linear range of detection.

## *P. falciparum* growth following heat stress when cultured in the presence of furosemide or with reduced nutrient medium

NF54 DiCre parasites were synchronised to a 2 hr invasion window and 1% parasitaemia before replacing the medium with either complete medium (as described above), complete medium with 150 µM furosemide or reduced nutrient medium. The composition of the reduced nutrient media was from *Molina et al., 2025* and is detailed in *Supplementary file 7*. A DMSO vehicle control was added to the complete medium and reduced nutrient medium. Between 16 and 24 hpi, iRBCs were either

heat stressed at 40 °C or maintained at 37 °C and parasitaemia was determined in the subsequent cycle at 72 hpi by flow cytometry. Following heat stress, cultures were maintained at 37 °C.

## NanoLuciferase *P. falciparum* export assay

NanoLuciferase (NanoLuc) expressing transgenic parasites were synchronised to a two-hour invasion window (1–3% parasitaemia, 1% haematocrit). To gather samples to measure activity, a 600 µL aliquot of parasite culture was pelleted (3000 × *g*, 1 min) and resuspended in 1 mL of PBS. This solution was then further divided into three 325 µL aliquots. These three aliquots were then pelleted, and the supernatant removed. The pellets were either resuspended in 200 µL of 0.1% NP40 in PBS, 0.03% Saponin (Sigma) in PBS or three haematolytic units (HUs) of Equinatoxin II (a kind gift from Mike Blackman) in PBS. These solutions were then incubated at 37 °C for 10 min. Following incubation, the samples were centrifuged (10,000 × *g* for 5 min), and 100 µL of the supernatant was collected without disturbing the pellet. The buffer composition of each of the fractions was normalised by the addition of 100 µL of the other two buffers. Therefore, each condition had a final volume of 300 µL and contained 1 HU of Equinatoxin II, 0.01% Saponin and 0.033% NP40 in PBS. These fractions were then kept at –20 °C until required.

To measure NanoLuc activity, fractions were first thawed on ice. 100 µL was transferred into a white opaque 96-well plate (Corning). Technical triplicates were typically measured in three different wells. NanoGlo substrate provided in the Nano-Glo Luciferase Assay System (Promega) was diluted 1:500 in PBS. 100 µL of the substrate dilution was then added to each well for a final substrate concentration of 1:1000. Luminescence was then measured with 225 gain in a Cytation 96-well plate reader (Agilent, BioTek).

## Immunofluorescence assay

Thin blood smears of iRBC cultures on microscope slides were submerged in ice-cold methanol at –20 °C to fix the cells. Slides were then wrapped in lint-free tissue and stored with silica beads at –80 °C until required. For labelling, slides were thawed at 37 °C in the presence of silica beads for 20 min. Once dry, hydrophobic rings were drawn on the slides, one for each labelling condition. Slides were blocked with 3% BSA in PBS for 1 hr at room temperature. Cells were labelled with primary antibodies diluted in PBS with 1% BSA for at least 1 hr at room temperature, applying 25 µL of antibody solution to each hydrophobic ring. After three washes with PBS, secondary antibodies containing BSA and DAPI were added and incubated under the same conditions for 1 hr. Finally, slides were washed, mounted with ProLong Gold antifade reagent (Thermo Fisher, P36930) and covered with a coverslip. Primary antibodies included rat α-HA (3F10, Merck [11867423001], 1:250), Rabbit α-SBP1 (*Mesén-Ramírez et al., 2016*; a kind gift from Tobias Spielmann, 1:5000), and Mouse α-KAHRP (The European Malaria Reagent Repository, mAb 18.2, 1:000). After three final washes with PBS, coverslips were mounted over the stained cells with Prolong Gold antifade reagent (Invitrogen) containing DAPI and sealed with nail polish. Images were taken using a Ti-E Nikon microscope with a ×100 TIRF objective at room temperature equipped with a LED-illumination. Alternatively, higher magnification images were acquired on a VisiTech instant SIM (VTChapteriSIM) microscope using a ×150 oil-immersion objective and 1.5 µm z-axis steps. Image processing was performed in FIJI.

## Immunoblotting

Schizonts were enriched by Percoll density separation and lysed in 1% SDS RIPA buffer supplemented with 1 mM DTT. Soluble cell extracts were first separated by SDS-PAGE before transferring onto a nitrocellulose membrane (Bio-Rad) by Transblot Turbo. As a loading control, VersaBlot CF680nm (Biotium) was added to protein samples before SDS-PAGE to enable visualisation of total protein content on the subsequent blot. Membranes were blocked overnight at 4 °C in 2% BSA, 0.2% Tween-20 in PBS. After blocking, membranes were probed with primary antibody for 1 hr at room temperature, before three washes in 0.2% Tween-20 in PBS (PBS-T). Primary antibodies included rat α-HA (3F10, Merck (11867423001), 1:3000) and Streptavidin-HRP (Pierce (21130), 1:5000). Washed membranes were then probed with the appropriate secondary antibody HRP conjugate (1:10000, Abcam) for an hour at room temperature. Membranes were finally washed with PBS-T three more times before adding HRP substrate and buffer (Immobilon ECL Ultra, Sigma) was added and the blot imaged by ChemiDoc (Bio-Rad). The total protein stain was visualised at 680 nm.

## Sample preparation for phosphoproteomic profiling of *P. falciparum*

The protocol for generating and processing *P. falciparum* phosphoproteome samples was previously described in detail by *Davies et al., 2020*. Synchronised NF54 *P. falciparum* parasites (0–2 hpi) were cultured at 37 °C and subjected to either heat stress (39 °C from 16 to 24 hpi) or maintained at 37 °C. While not heat stressed, parasites were kept at 37 °C. iRBCs were enriched at 30 hpi using a 60% Percoll density gradient centrifugation and washed three times with PBS. Parasite pellets were lysed in 8 M urea in 50 mM HEPES (pH 8.5), supplemented with protease inhibitors (cOmplete Mini, EDTA-free, Roche) and phosphatase inhibitors (PhosSTOP, Roche). Lysates were sonicated (3×30 s, 30% duty cycle, on ice), clarified by centrifugation (13,000 × $g$, 30 min, 4 °C), and the supernatants were snap-frozen in liquid nitrogen and stored at −80 °C.

For phosphoproteomic analysis, a TMT 10-plex (Thermo Fisher Scientific) was used for wild-type parasites ±heat stress (two biological replicates per condition, using RBCs from independent donors). Two additional channels were used for uRBCs ±heat stress (N=1 each), and the remaining four channels were used for unrelated experiments. A second experiment used TMTpro 16-plex (Thermo Fisher Scientific) labelling across seven knockout lines (FIKK1, FIKK4.1, FIKK4.2, FIKK9s, FIKK10.1, FIKK10.2) ± heat stress, alongside wild-type and uRBC controls (N=1 per condition). These knockout strains were previously generated and validated by *Davies et al., 2020*. For the conditional knockout strains, gene disruption was induced by DiCre-mediated excision as described above, in the preceding intraerythrocytic cycle to ensure full loss of gene function. These strains, alongside the NF54 wild-type, were then resynchronised at the start of the cycle (0–2 hpi) prior to heat stress and sample collection.

## Reduction, alkylation, and protein digest of phosphoproteome samples

Once thawed, protein concentration was determined by a BCA protein assay (Pierce). 1 mg of each lysate was subsequently reduced with 10 mM DTT for 60 min at room temperature and alkylated in the dark with 20 mM iodoacetamide for 30 min at room temperature. Excess iodoacetamide was then quenched with 10 mM DTT. Lysates were diluted with 50 mM HEPES pH 8.5 to <2 M urea and each digested with 5 µg of LysC (WAKO) for 2–3 hr at 37 °C and then overnight with 30 µg of trypsin (Thermo Scientific) at 37 °C.

## Sep-Pak desalting

The samples were acidified with trifluoroacetic acid (TFA; Thermo Fisher Scientific) to a final concentration of 0.4% (v/v) and left on ice for 10 min. All insoluble material was removed by centrifugation (1600 × $g$, 10 min, 4 °C), and the supernatants were desalted using Sep-Pak Lite C18 cartridges (Waters) in conjunction with a vacuum manifold. Columns were initially washed with 3 mL of acetonitrile, conditioned with 1 mL of 50% acetonitrile and 0.5% acetic acid in $H_2O$, and then equilibrated with 3 mL of 0.1% TFA in $H_2O$. The acidified samples were loaded, desalted with 3 mL of 0.1% TFA in $H_2O$, washed with 1 mL of 0.5% acetic acid in $H_2O$, and finally eluted with 1.2 mL of 50% acetonitrile and 0.5% acetic acid in $H_2O$. Each sample was subsequently dried by vacuum centrifugation.

## TMT labelling

Samples were dissolved in 1 mL of 50 mM Na-HEPES pH 8.5 and 30% anhydrous acetonitrile (v/v) and labelled with the respective TMT reagents (Thermo Fisher Scientific). Both TMT-10plex (2.4 mg reagent/1 mg sample) and TMT-16plex (3 mg reagent/1 mg sample) were used in two separate experiments. Labelling reactions were carried out for one hour at room temperature, followed by quenching with 0.3% hydroxylamine for 15 min at room temperature and sample acidification (pH ~2) with formic acid. After verification of labelling efficiency via mass spectrometry, the lysates were mixed in a 1:1 ratio, vacuum dried and desalted on Sep-Pak C18 cartridges.

## Phosphopeptide enrichment and high pH sample fractionation

Phosphopeptides were sequentially enriched using the High Select SMOAC protocol. In the first step, a High Select TiO2 phosphopeptide enrichment kit (Thermo Fisher Scientific) was used according to the manufacturer instructions.

The supernatant from the initial TiO₂ enrichment was vacuum dried, and a second enrichment was performed using the High Select Fe-NTA phosphopeptide enrichment kit (Thermo Fisher Scientific), following the manufacturer's instructions. The flow-through, containing non-phosphorylated peptides

(representing the total proteome), was retained and stored at −80 °C. Phosphopeptide eluates from $TiO_2$ and Fe-NTA enrichment were pooled and fractionated using the Pierce High pH Reversed-Phase Peptide Fractionation Kit (Thermo Fisher Scientific), following the manufacturer's instructions for unlabelled native peptides. The resulting eight fractions were subsequently dried under vacuum to prepare for LC-MS/MS.

## LC-MS/MS (phosphoproteomics) and data analysis

TMT-10plex: samples were resuspended in 0.1% TFA and loaded onto a 50-cm EasySpray PepMap column (75 µm inner diameter, 2 µm particle size; Thermo Fisher Scientific) fitted with an integrated electrospray emitter. Reverse-phase chromatography was performed using an RSLC nano U3000 system (Thermo Fisher Scientific) with a binary buffer system: solvent A consisted of 0.1% formic acid and 5% dimethyl sulfoxide (DMSO) in water, while solvent B contained 80% acetonitrile, 0.1% formic acid, and 5% DMSO. Chromatographic separation was carried out at a flow rate of 275 nL/min. Samples were subjected to a linear gradient of 2–25% solvent B over 120 min, followed by 25–40% B over 25 min, 40–90% B in 5 min, and finally 90% B over 10 min. The total runtime of the LC separation was 180 min, including a 20-min column conditioning. The nanoLC system was coupled to an Orbitrap Lumos mass spectrometer via an EasySpray nano source (Thermo Fisher Scientific). The Orbitrap Lumos was operated in data-dependent acquisition mode using Xcalibur software. For MS2 acquisition, higher-energy collisional dissociation (HCD) MS/MS scans were recorded at a resolution of 60,000 following MS1 survey scans acquired at a resolution of 120,000. The MS1 ion target was set to $4 \times 10^5$, and the MS2 target to $1 \times 10^{55}$. Fragmentation was performed using a 'Top Speed' acquisition strategy with a 3 s cycle time and a dynamic exclusion window of 45 s. The maximum ion injection time for MS2 scans was 105ms, and the normalised HCD collision energy was set to 38. For MS3 acquisition, collision-induced dissociation (CID) MS2 scans were acquired at a resolution of 30,000, following an MS1 scan with the same parameters as above. The MS2 ion target was $5 \times 10^4$, and multistage activation of the neutral loss ($H_3PO_4$) was enabled. The maximum injection time for MS2 scans was 60 ms, with a CID collision energy of 35. HCD MS3 scans were acquired at a resolution of 60,000, using synchronous precursor selection to include up to ten MS2 fragment ions. The MS3 ion target was $1 \times 10^5$, with a maximum injection time of 105 ms and a collision energy of 65.

TMT-16plex: samples were resuspended in 0.1% TFA, 2% acetonitrile and loaded onto a 50-cm EasySpray PepMap column (75 µm inner diameter, 2 µm particle size; Thermo Fisher Scientific) fitted with an integrated electrospray emitter. Reverse-phase chromatography was performed using an RSLC nano U3000 system (Thermo Fisher Scientific) with a binary buffer system: solvent A consisted of 0.1% formic acid and 5% dimethyl sulfoxide (DMSO) in water, while solvent B contained 80% acetonitrile, 0.1% formic acid and 5% DMSO. Chromatographic separation was carried out at a flow rate of 250 nL/min. Samples were subjected to a linear gradient of 2–35% solvent B over 120 min, followed by 35–45% B over 25 min, 45–95% B in 5 min and finally 95% B over 10 min. The total runtime of the LC separation was 180 min, including a 20-min column conditioning. The nanoLC system was coupled to an Orbitrap Lumos mass spectrometer via an EasySpray nano source (Thermo Fisher Scientific). The Orbitrap Lumos was operated in data-dependent acquisition mode using Xcalibur software. For MS2 acquisition, higher-energy collisional dissociation (HCD) MS/MS scans were recorded at a resolution of 50,000 following MS1 survey scans acquired at a resolution of 120,000. The MS1 ion target was set to $4 \times 10^5$, and the MS2 target to $1 \times 10^5$. Fragmentation was performed using a 'Top Speed' acquisition strategy with a 3 s cycle time and a dynamic exclusion window of 45 s. The maximum ion injection time for MS2 scans was 120 ms, and the normalised HCD collision energy was set to 35. For MS3 acquisition, collision-induced dissociation (CID) MS2 scans were acquired at a resolution of 30,000, following an MS1 scan with the same parameters as above. The MS2 ion target was $5 \times 10^4$, and multistage activation of the neutral loss ($H_3PO_4$) was enabled. The maximum injection time for MS2 scans was 60 ms, with a CID collision energy of 30. HCD MS3 scans were acquired at a resolution of 50,000, using synchronous precursor selection to include up to 10 MS2 fragment ions. The MS3 ion target was $1 \times 10^5$, with a maximum injection time of 120 ms and a collision energy of 55.

Raw mass spectrometry data were processed using MaxQuant *Cox and Mann, 2008* v1.6.2.10 (TMT-10plex) and v1.6.12.0 (TMT-16plex), with peptide identification carried out via the Andromeda *Cox et al., 2011* search engine. Spectra were searched against *Plasmodium falciparum* *Aurrecoechea et al., 2009* and *Homo sapiens* *UniProt, 2019* proteomes. TMT-based quantification was performed

using the built-in 'reporter ion MS2 or MS3' algorithm in MaxQuant, with the reporter ion mass tolerance set to 0.003 Da. Carbamidomethylation of cysteine was set as a fixed modification, while variable modifications included methionine oxidation, protein N-terminal acetylation, and phosphorylation (S, T, Y). Enzyme specificity was set to trypsin, allowing up to two missed cleavages. The precursor mass tolerance was set to 20 ppm for the first search (used for mass recalibration) and 4.5 ppm for the main search. The 'match between runs' option was enabled (time window of 0.7 min) for fractionated samples. All datasets were filtered using posterior error probability (PEP) to achieve a 1% false discovery rate at the protein, peptide, and site levels.

Mass spectrometry datasets were imported into Perseus v1.5.0.9 *Tyanova et al., 2016* and filtered to exclude common contaminants as well as identifications derived from reverse decoy sequences. To generate a list of all quantified phosphorylation sites, reporter intensities were filtered to retain entries with at least one valid value. TMT reporter intensity values were $log_2$ transformed and normalised by median subtraction. Fold changes between conditions were then calculated for each experiment. Normalisation was performed in two steps. First, for each phosphosite, the mean $log_2$ intensity across all samples was subtracted from individual values to centre the data per site. Second, the median $log_2$ intensity of each sample was subtracted from all values in that sample to correct for global sample-level intensity differences *Love et al., 2014*. For each phosphosite quantified across all samples in a given experimental run, the mean $log_2$ intensity was calculated (i.e. the row mean). This mean was subtracted from each $log_2$ intensity value within that row. A median was then calculated for each column in the resulting matrix to determine a scaling factor for each sample. These median values were subtracted from the corresponding original $log_2$ intensity values within each column to obtain the final normalised intensities. The phosphoproteomic mass spectrometry data have been deposited to the ProteomeXchange Consortium via the PRIDE *Perez-Riverol et al., 2019* partner repository with the dataset identifier PXD073843.

## Proteome analysis

Approximately 100 µg of the non-phosphorylated peptide flow-through was subjected to high-pH reversed-phase fractionation using the Pierce High pH Reversed-Phase Peptide Fractionation Kit (Thermo Fisher Scientific), following the manufacturer's protocol for TMT-labelled samples with an additional 5% acetonitrile wash. Fractions were dried by vacuum centrifugation, stored at –80 °C, and subsequently analysed by LC-MS/MS on an Orbitrap Lumos mass spectrometer using data-dependent MS2 acquisition method as described above for TMT-10plex. Raw mass spectrometry data were processed using MaxQuant v1.6.2.10 *Cox and Mann, 2008* as described above with the following variable modifications: methionine oxidation, protein N-terminal acetylation, deamidation (NQ). Data were then imported into Perseus v1.5.0.9 (*Tyanova et al., 2016*) and filtered to exclude common contaminants as well as identifications derived from reverse decoy sequences and only identified by site. TMT reporter intensity values were $log_2$ transformed, normalised by median subtraction and filtered for one valid value to generate a list of quantified proteins.

## FIKK10.2-TurboID proxiome labelling

The protocol for TurboID-based proximity labelling and biotinylated peptide enrichment was previously described in detail by *Davies et al., 2023*. TurboID-expressing parasites were cultured in biotin-free medium for two cycles prior to labelling. Parasites were tightly synchronised to a 2 hr invasion window (>10% parasitaemia, 2% haematocrit in 300 mL total volume), and cultures were split into three experimental conditions, each performed in triplicate using RBCs from independent donors (N = 3, 9 samples total).

Labelling was initiated by resuspending cells in medium containing 100 µM D-biotin (bioAPE); while not undergoing labelling, all cultures were maintained in biotin-free medium. Parasites were subjected to one of three total labelling conditions: continuous biotin labelling for 40 hr at 37 °C, or an 8 hr biotin pulse from 16 to 24 hpi at either 37 °C or 39 °C (heat stress). Outside these windows, biotin was removed by pelleting cultures and resuspending them in fresh biotin-free medium. At 40 hpi, samples were enriched by 60% Percoll gradient centrifugation and washed five times with PBS. Cells were lysed in ice-cold 8 M urea in 50 mM HEPES (pH 8.0), supplemented with cOmplete Mini, EDTA-free protease inhibitors (Roche). Lysates were sonicated (30% duty cycle, 3×30 second bursts

on ice), clarified by centrifugation (13,000×g, 30 minutes, 4 °C), snap-frozen in liquid nitrogen, and stored at −80 °C.

## Reduction, alkylation, and protein digest of biotinylated peptides

Parasite lysates were thawed and centrifuged to pellet insoluble material (15 min, 4 °C, 21,000 × g). Protein content in the supernatant was quantified using the BCA Protein Assay Kit (Pierce). An equal amount of parasite material (4 mg per sample) was taken, and protein concentrations were equalised using 8 M urea in 50 mM HEPES. Oxidised cysteine residues were reduced with 5 mM dithiothreitol (DTT) at room temperature for 60 min, followed by alkylation of the reduced cysteines with 10 mM iodoacetamide in the dark at room temperature for 30 min. For protein digestion, samples were diluted fourfold with 50 mM HEPES to reduce the urea concentration to below 2 M. Mass spectrometry-grade trypsin was added at a 1:50 enzyme-to-protein ratio and incubated at 37 °C overnight (16 hr). To terminate digestion, samples were cooled on ice for 10 min and acidified with TFA to a final concentration of ~0.4% (v/v), incubating on ice for a further 10 min. Samples were centrifuged at 21,000 × g for 15 min at 4 °C to remove insoluble material. These peptides were then Sep-Pak desalted and vacuum-dried as previously described.

## Charging protein G agarose beads with anti-biotin antibodies

For each sample, 60 µL of a 50% protein-G agarose bead slurry (Pierce #20398) was used and washed three times with 5–10 bead volumes of BioSITe buffer (*Kim et al., 2018*) comprising 50 mM Tris, 150 mM NaCl, and 0.5% Triton X-100 (pH 7.2–7.5 at 4 °C). Between washes, the beads were pelleted by centrifugation at 1500 × g for 2 min at 4 °C. Anti-biotin antibodies (a mixture of 30 µg each of Bethyl Laboratories #150–109 A and Abcam #Ab53494) were added per 60 µL of bead slurry, brought up to 2 mL with BioSITe buffer, and incubated overnight at 4 °C with rotation. The beads were then washed three times with BioSITe buffer, adjusted to approximately a 50% slurry using the same buffer, and aliquoted into 60 µL portions for each sample.

## Biotinylated peptide immunoprecipitation

Dried peptide samples were reconstituted in 1.5 mL of BioSITe buffer with vortexing to facilitate dissolution. The pH was adjusted to between 7 and 7.5 on ice using 1–5 µL of 10 M NaOH. Samples were then centrifuged at 21,000 × g for 10 min at 4 °C to remove any insoluble material. Peptide concentrations were determined using the peptide BCA assay kit (Pierce, #23275). Equal peptide amounts from each sample were added to the anti-biotin antibody-conjugated bead slurry (~60 µL per sample) and incubated with rotation for 2 hr at 4 °C. Following incubation, beads were pelleted by centrifugation (1500 × g, 2 min, 4 °C) and sequentially washed with 3 × 0.5 mL BioSITe buffer, 1 × 0.5 mL of 50 mM Tris, and 3 × 1 mL of mass spectrometry-grade water. Biotinylated peptides were eluted by treating the beads with 4 × 50 µL of 0.2% TFA, and eluates were stored at −80 °C.

## LC-MS/MS (TurboID biotinylated peptide pulldown) and data analysis

Samples were loaded onto Evotips (according to the manufacturer's instructions). Following a wash with aqueous acidic buffer (0.1% formic acid in water), samples were loaded onto an Evosep One system coupled to an Orbitrap Fusion Lumos (Thermo Fisher Scientific). The Evosep One was fitted with a 15 cm column (PepSep), and a predefined gradient for a 44 min method was employed. The Orbitrap Lumos was operated in data-dependent mode (1 s cycle time), acquiring IT HCD MS/MS scans in rapid mode after an OT MS1 survey scan ($R$=60,000). The MS1 target was 4E5 ions, whereas the MS2 target was 1E4 ions. The maximum ion injection time utilised for MS2 scans was 300 ms, the HCD normalised collision energy was set at 32, and the dynamic exclusion was set at 15 s.

Acquired raw files were processed with MaxQuant v1.6.2.10 (*Cox and Mann, 2008*). Peptides were identified from the MS/MS spectra searched against *P. falciparum* (*Aurrecoechea et al., 2009*) and *Homo sapiens* (*UniProt, 2019*) proteomes using Andromeda (*Cox et al., 2011*) search engine. Biotin (K), Oxidation (M), Acetyl (Protein N-term), and Phospho (STY) were selected as variable modifications, whereas Carbamidomethyl (C) was selected as a fixed modification. The enzyme specificity was set to trypsin with a maximum of three missed cleavages. Minimal peptide length was set at six amino acids. Biotinylated peptide search in MaxQuant was enabled by defining a biotin adduct (+226.0776) on lysine residues as well as its three diagnostic ions: fragmented biotin (m/z 227.0849), immonium

ion harbouring biotin with a loss of NH3 (m/z 310.1584), and an immonium ion harbouring biotin (m/z 327.1849). The precursor mass tolerance was set to 20 ppm for the first search (used for mass re-calibration) and to 4.5 ppm for the main search. The datasets were filtered on posterior error probability (PEP) to achieve a 1% false discovery rate on protein, peptide, and site level. Other parameters were used as pre-set in the software.

MaxQuant output files were processed with Perseus, v1.5.0.9 (*Tyanova et al., 2016*), the data were filtered to exclude common contaminants as well as identifications derived from reverse decoy sequences. Peptide intensities were log₂-transformed, and peptides were filtered to retain those detected in two out of three replicates within at least one condition. Valid peptides were assigned for each group as being present in at least two of three replicate conditions. Valid proteins were defined for each condition as having at least two valid peptides. Valid peptides and proteins from each condition were used to generate area-proportional Venn diagrams using the BioVenn tool (*Hulsen et al., 2008*). The TurboID mass spectrometry data have been deposited to the ProteomeXchange Consortium via the PRIDE (*Perez-Riverol et al., 2019*) partner repository with the dataset identifier PXD073890.

## Statistics and reproducibility

Data were analysed using GraphPad Prism v10 and details of the analyses are provided in each figure legend. Where specified, data were log-transformed prior to statistical testing. Percentage data constrained between 0 and 100% were first converted to proportions and then Logit transformed. Fold change data were Log10 transformed as indicated. Multiple comparisons were corrected for false discovery rate (FDR) using the method of Benjamini, Krieger, and Yekutieli. Where FDR correction was applied, adjusted p values (Q values) are reported. Statistical significance was defined as follows: ns not significant, *p<0.05, **p<0.01, ***p<0.001, ****p<0.0001.

## Acknowledgements

We thank members of the Treeck lab for critical discussions and additionally Dr. Christian Gnann, Dr. Franziska Hildebrandt and Miss Ana Matias for their feedback on this manuscript. We would like to thank members of the Sateriale, Blackman and Bernabeu lab groups for critical discussions throughout the project. We also thank the Crick Science Technology Platforms (Proteomics, Flow Cytometry, Media Preparation and Light Microscopy) for their outstanding technical support. Special thanks to PlasmoDB for providing a critical resource. We thank Colin Sutherland for providing the HL2208 *P. falciparum* strain, Lars Hviid for the α-VAR2CSA (PAM7.5) antibody, Tobias Spielmann for the α-SBP1 antibody, and the European Malaria Reagent Repository for the α-KAHRP antibody. MT is supported by funding from the ERC (Grant Number 10144428), which also provided funding to HB, GF and DA, The Francis Crick Institute (Grant Numbers CC2132 & CR2023/030/2132), which also provided funding to HB. The Francis Crick Institute Science Technology Platforms (Grant Number CC0199) and the FCT (Fundación para a Ciencia e Tecnologia (Grant Number 2023.06167.CEECIND)). SDN is funded by an Early-Career Award Wellcome Trust grant (225686/Z/22/Z). GF is funded by the FCT through a PhD fellowship (Grant Number 2024.02695.BD). The Francis Crick Institute and its Science Technology platforms receive core funding from Cancer Research UK, the UK Medical Research Council and the Wellcome Trust.

## Additional information

### Competing interests

Moritz Treeck: Reviewing editor, eLife. The other authors declare that no competing interests exist.

### Funding

| Funder | Grant reference number | Author |
| --- | --- | --- |
| European Commission | 10144428 | Hugo Belda<br>David Anaguano<br>Moritz Treeck |

| Funder | Grant reference number | Author |
| --- | --- | --- |
| The Francis Crick Institute | CC2132 | Malgorzata Broncel<br>Gwendolin Fuchs<br>Moritz Treeck |
| The Francis Crick Institute | CR2023/030/2132 | Moritz Treeck |
| Fundação para a Ciência e a Tecnologia | 2023.06167.CEECIND | Moritz Treeck |
| Wellcome Trust | 10.35802/225686 | Stephanie D Nofal |
| Fundação para a Ciência e a Tecnologia | 2024.02695.BD | Gwendolin Fuchs |
| The Francis Crick Institute | CC0199 | Moritz Treeck |

The funders had no role in study design, data collection and interpretation, or the decision to submit the work for publication. For the purpose of Open Access, the authors have applied a CC BY public copyright license to any Author Accepted Manuscript version arising from this submission.

## Author contributions

David Jones, Conceptualization, Formal analysis, Investigation, Visualization, Methodology, Writing – original draft, Writing – review and editing; Hugo Belda, Malgorzata Broncel, Gwendolin Fuchs, David Anaguano, Stephanie D Nofal, Formal analysis, Investigation, Writing – review and editing; Moritz Treeck, Conceptualization, Formal analysis, Supervision, Funding acquisition, Methodology, Writing – original draft, Project administration

## Author ORCIDs

David Jones ⓘ https://orcid.org/0000-0002-4789-203X
Hugo Belda ⓘ https://orcid.org/0000-0001-8628-9455
Malgorzata Broncel ⓘ https://orcid.org/0000-0003-2991-3500
Gwendolin Fuchs ⓘ https://orcid.org/0000-0001-9294-6984
David Anaguano ⓘ https://orcid.org/0000-0001-7384-8377
Stephanie D Nofal ⓘ https://orcid.org/0000-0003-1415-3369
Moritz Treeck ⓘ https://orcid.org/0000-0002-9727-6657

Reviewer #1 (Public review): https://doi.org/10.7554/eLife.107860.3.sa1
Reviewer #2 (Public review): https://doi.org/10.7554/eLife.107860.3.sa2
Reviewer #3 (Public review): https://doi.org/10.7554/eLife.107860.3.sa3
Author response https://doi.org/10.7554/eLife.107860.3.sa4

# Additional files

## Supplementary files

Supplementary file 1. Heat stress phosphoproteome and proteome data from the 10-plex TMT from NF54::DiCre (wild type). Includes *P. falciparum* and *H. sapiens* peptides and proteins.

Supplementary file 2. P. *falciparum* phosphosites that are more phosphorylated following heat stress.

Supplementary file 3. Heat stress phosphoproteome and proteome data from the 16-plex TMT with FIKK conditional knockouts. Includes *P. falciparum* and *H. sapiens* peptides and proteins.

Supplementary file 4. FIKK10.2-TurboID proximal biotinylated proteins identified under various conditions, including the presence or absence of biotin and heat stress.

Supplementary file 5. FIKK10.2-TurboID proximal biotinylated proteins identified in the presence of biotin and in the absence of heat stress.

Supplementary file 6. Primer sequences and binding sites used for validation of transgenic *P. falciparum* strains.

Supplementary file 7. Composition of reduced nutrient media.

MDAR checklist

## Data availability

All data used in this study is provided within the manuscript as supplementary files or source data. Proteome raw data has been provided through the PRIDE respository (PXD073843). Cell lines and reagents can be obtained upon request.

The following datasets were generated:

| Author(s) | Year | Dataset title | Dataset URL | Database and Identifier |
|---|---|---|---|---|
| Jones D | 2026 | Phosphoproteomics of *Plasmodium falciparum* infected red blood cells under Heat Stress | https://www.ebi.ac.uk/pride/archive/projects/PXD073843 | PRIDE, PXD073843 |
| Jones D | 2026 | TurboID Proximity Profiling of FIKK10.2 (PF3D7_1039000) in *Plasmodium falciparum*–Infected Red Blood Cells During Febrile Heat Stress | https://www.ebi.ac.uk/pride/archive/projects/PXD073890 | PRIDE, PXD073890 |

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
