## [Editor Report · eLife Assessment]

This **important** study provides **compelling** evidence that fever-like temperatures enhance the export of *Plasmodium falciparum* transmembrane proteins, including the cytoadherence protein PfEMP1 and the nutrient channel PSAC, to the red blood cell surface, thereby increasing cytoadhesion. Using rigorous and well-controlled experiments, the authors **convincingly** demonstrate that this effect results from accelerated protein trafficking rather than changes in protein production or parasite development. These findings significantly advance our understanding of parasite virulence mechanisms and offer insights into how febrile episodes may exacerbate malaria severity.

---

## [Referee Report · Reviewer #1 (Public review)]

Summary:

The manuscript from Jones and colleagues investigates a previously described phenomenon in which *P. falciparum* malaria parasites display increased trafficking of proteins displayed on the surface of infected RBCs as well as increased cytoadherence in response to febrile temperatures. While this parasite response was previously described, it was not uniformly accepted, and conflicting reports can be found in the literature. This variability likely arises due to differences in the methods employed and the degree of temperature increase that the parasites were exposed to. Here the authors are very careful to employ a temperature shift that likely reflects what is happening in infected humans and that they demonstrate is not detrimental to parasite viability or replication. In addition, they go on to investigate what steps in protein trafficking are affected by exposure to increased temperature and show that the effect is not specific to PfEMP1 but rather likely affects all transmembrane domain containing proteins that are trafficked to the RBC. They also detect increased rates of phosphorylation of trafficked proteins, consistent with overall increased protein export.

Strengths:

The authors used a relatively mild increase in temperature (39 degrees) that they demonstrate is not detrimental to parasite viability or replication. This enabled them to avoid potential complications of more severe heat shock that might have affected previously published studies. They employed a clever method of fractionation of RBCs infected with a var2csa-nanoluc fusion protein expressing parasite line to determine which step in the export pathway was likely accelerating in response to increased temperature. This enabled them to determine that export across the PVM is being affected. They also explored changes in phosphorylation of exported proteins and demonstrated that the effect is not limited to PfEMP1 but appears to affect numerous (or potentially all) exported transmembrane domain containing proteins.

Impact and conclusions:

The study shows that protein export, including PfEMP1 and PSAC, are accelerated in response to mild heat shock. This has implications for disease severity as well as our understanding of protein trafficking in these unique organisms. There is increasing interest in asymptomatic infections, which have been proposed to be a major reservoir for transmission and generally are not associated with fever. It will be interesting to consider whether reduced (or slower) trafficking of these proteins has a selective advantage for parasites in asymptomatic infections.

---

## [Referee Report · Reviewer #2 (Public review)]

This manuscript describes experiments characterising how malaria parasites respond to physiologically relevant heat-shock conditions. The authors show, quite convincingly, that moderate heat-shock appears to increase cytoadherance, likely by increasing trafficking of surface proteins involved in this process.

While generally of a high quality and including a lot of data, I have a few small questions and comments, mainly regarding data interpretation.

(1) The authors use sorbitol lysis as a proxy for trafficking of PSAC components. This is a very roundabout way of doing things and does not, I think, really show what they claim. There could be a myriad of other reasons for this increased activity (indeed, the authors note potential PSAC activation under these conditions). One further reason could be a difference in the membrane stability following heat shock, which may affect sorbitol uptake, or the fragility of the erythrocytes to hypotonic shock. I really suggest that the authors stick to what they show (increased PSAC) without trying to use this as evidence for increased trafficking of a number of non-specified proteins that they cannot follow directly.

(2) Supplementary Figure 6C/D: The KAHRP signal does not look like it should. In fact, it doesn't look like anything specific. The HSP70-X signal is also blurry and overexposed. These pictures cannot be used to justify the authors' statements about a lack of colocalisation in any way.

(3) Figure 6: This experiment confuses me. The authors purport to fractionate proteins using differential lysis, but the proteins they detect are supposed to be transmembrane proteins and thus should always be found associated with the pellet, whether lysis is done using equinatoxin or saponin. Have they discovered a currently unknown trafficking pathway to tell us about? Whilst there is a lot of discussion about the trafficking pathways for TM proteins through the host cell, a number of studies have shown that these proteins are generally found in a membrane-bound state. The authors should elaborate, or choose an experiment that is capable of showing compartment-specific localisation of membrane-bound proteins (protease protection, for example).

(4) The red blood cell contains, in addition to HSP70-X, a number of human HSPs (HSP70 and HSP90 are significant in this current case). As the name suggests, these proteins non-specifically shield exposed hydrophobic domains revealed upon partial protein unfolding following thermal insult. I would thus have expected to find significantly more enrichment following heat shock, but this is not the case. Is it possible that the physiological heat shock conditions used in this current study are not high enough to cause a real heat shock?

Comments on Revision:

Although in any study there are going to be residual weaknesses, this reviewer is happy to see that the authors have gone to lengths to address many of my main concerns, and also those of other reviewers.

---

## [Referee Report · Reviewer #3 (Public review)]

Summary:

In this paper it is established that high fever-like 39oC temperatures cause parasite infected red blood cells become stickier. It is thought that high temperatures might help the spleen to destroy parasite infected cells, so they become stickier to remain trapping in blood vessels, so they stop passing through the spleen.

Strengths:

The strength of this research is that it shows that fever-like temperatures can cause parasite infected red blood cells to stick to surfaces designed to mimic the walls of small blood vessels. In a natural infection this would cause parasite infected red blood cells to stop circulating through the spleen where the parasites would be destroyed by the immune system. It is thought that fevers could lead to infected red blood cells becoming stiffer and therefore more easily destroyed in the spleen. Parasites respond to fevers by making their red blood cells stickier, so they stop flowing around the body and into the spleen. The experiments here prove fever temperatures increase the export of Velcro-like sticky proteins onto the surface of the infected red blood cells and are very thorough and convincing.

Weaknesses:

Minor weaknesses in the original version have now been satisfactorily addressed with additional work which is very convincing.

---

## [Author Response]

The following is the authors’ response to the original reviews.

**eLife Assessment**
This important study provides compelling evidence that fever-like temperatures enhance the export of *Plasmodium falciparum* transmembrane proteins, including the cytoadherence protein PfEMP1 and the nutrient channel PSAC, to the red blood cell surface, thereby increasing cytoadhesion. Using rigorous and well-controlled experiments, the authors convincingly demonstrate that this effect results from accelerated protein trafficking rather than changes in protein production or parasite development. These findings significantly advance our understanding of parasite virulence mechanisms and offer insights into how febrile episodes may exacerbate malaria severity.

We thank all reviewers for their constructive feedback on our manuscript.

We believe we have addressed all the questions in the rebuttal below in writing, including planned experiments we will perform to strengthen the conclusions of the manuscript.

**Public Reviews:**

**Reviewer #1 (Public review):**
Summary:This manuscript from Jones and colleagues investigates a previously described phenomenon in which *P. falciparum* malaria parasites display increased trafficking of proteins displayed on the surface of infected RBCs, as well as increased cytoadherence in response to febrile temperatures. While this parasite response was previously described, it was not uniformly accepted, and conflicting reports can be found in the literature. This variability likely arises due to differences in the methods employed and the degree of temperature increase to which the parasites were exposed. Here, the authors are very careful to employ a temperature shift that likely reflects what is happening in infected humans and that they demonstrate is not detrimental to parasite viability or replication. In addition, they go on to investigate what steps in protein trafficking are affected by exposure to increased temperature and show that the effect is not specific to PfEMP1 but rather likely affects all transmembrane domain-containing proteins that are trafficked to the RBC. They also detect increased rates of phosphorylation of trafficked proteins, consistent with overall increased protein export.Strengths:The authors used a relatively mild increase in temperature (39 degrees), which they demonstrate is not detrimental to parasite viability or replication. This enabled them to avoid potential complications of a more severe heat shock that might have affected previously published studies. They employed a clever method of fractionation of RBCs infected with a var2csa-nanoluc fusion protein expressing parasite line to determine which step in the export pathway was likely accelerating in response to increased temperature. This enabled them to determine that export across the PVM is being affected. They also explored changes in phosphorylation of exported proteins and demonstrated that the effect is not limited to PfEMP1 but appears to affect numerous (or potentially all) exported transmembrane domain-containing proteins.Weaknesses:All the experiments investigating changes resulting from increased temperature were conducted after an increase in temperature from 16 to 24 hours, with sampling or assays conducted at the 24 hr mark. While this provided consistency throughout the study, this is a time point relatively early in the export of proteins to the RBC surface, as shown in Figure 1E. At 24 hrs, only approximately 50% of wildtype parasites are positive for PfEMP1, while at 32 hrs this approaches 80%. Since the authors only checked the effect of heat stress at 24 hrs, it is not possible to determine if the changes they observe reflect an overall increase in protein trafficking or instead a shift to earlier (or an accelerated) trafficking. In other words, if a second time point had been considered (for example, 32 hrs or later), would the parasites grown in the absence of heat stress catch up?

We did not assess cytoadhesion at later stages, but in the supplementary figures we show that at 40 hours post infection both heat stress and control conditions have comparable proportions of VAR2CSA-positive iRBCs, whilst they differ at 24h. This is true for the DMSO (control wildtype resembling) HA-tagged lines of HSP70x and PF3D7_072500 (Supplementary Figures 9 and 12 respectively). In the light that protein levels appear not changed, we conclude that trafficking is accelerated during these earlier timepoints, but remains comparable at later stages. This would still increase the overall bound parasite mass as parasites start to adhere earlier during or after a heat stress.

**Reviewer #2 (Public review):**
This manuscript describes experiments characterising how malaria parasites respond to physiologically relevant heat-shock conditions. The authors show, quite convincingly, that moderate heat-shock appears to increase cytoadherance, likely by increasing trafficking of surface proteins involved in this process.While generally of a high quality and including a lot of data, I have a few small questions and comments, mainly regarding data interpretation.(1) The authors use sorbitol lysis as a proxy for trafficking of PSAC components. This is a very roundabout way of doing things and does not, I think, really show what they claim. There could be a myriad of other reasons for this increased activity (indeed, the authors note potential PSAC activation under these conditions). One further reason could be a difference in the membrane stability following heat shock, which may affect sorbitol uptake, or the fragility of the erythrocytes to hypotonic shock. I really suggest that the authors stick to what they show (increased PSAC) without trying to use this as evidence for increased trafficking of a number of non-specified proteins that they cannot follow directly.

This is a valid point, however, uninfected RBCs do not lyse following heat stress, nor do much younger iRBCs, indicating that the observed effect is specific to infected RBCs at a defined stage. The sorbitol sensitivity assay is performed at 37°C under normal conditions after cells are returned to non–heat stress temperatures, so the effect is not due to transient changes in membrane permeability at elevated temperature.

Planned experiment: However, to increase the strength of our conclusions and further test our hypothesis, we will perform sorbitol sensitivity assays on >20 hours post infection iRBCs following heat stress in the presence and absence of furosemide, a PSAC inhibitor. If iRBC lysis is abolished with furosemide present, this would confirm that the effect is PSAC-dependent. However, the effect could also possibly be due to altered PSAC activity during heat stress which is maintained at lower temperatures, as outlined in the discussion.

New Results:

We performed sorbitol sensitivity assays on >20 hours post-infection iRBCs following heat stress in the presence and absence of the PSAC inhibitor furosemide. These additional experiments were added to the supplementary figures (Supplementary Figure 3). Importantly, sorbitol-mediated lysis of iRBCs, with or without prior heat stress, was reduced when furosemide was present, demonstrating that the observed effect is likely PSAC-dependent. We also observed that uninfected RBCs did not lyse with sorbitol, regardless of heat stress, confirming that the effect is specific to infected cells.

(2) Supplementary Figure 6C/D: The KAHRP signal does not look like it should. In fact, it doesn't look like anything specific. The HSP70-X signal is also blurry and overexposed. These pictures cannot be used to justify the authors' statements about a lack of colocalisation in any way.

Planned experiment: We agree that the IFAs are not the best as presented and will include better quality supplementary images in a revised version.

New Results:

Immunofluorescence microscopy, including the localisation of the two HA-tagged proteins (PF3D7_1039000 and PF3D7_0702500), has been repeated and higher-quality images are now included in the updated manuscript (Supplementary Figures 9 and 11). These images include co-staining with the *P. falciparum* proteins KAHRP and SPB1 to assess possible co-localisations. Furthermore, following the reviewer’s suggestion, we have softened the statement regarding PF3D7_1039000-HA to better reflect the data, changing “...does not colocalise” to “...does not strongly colocalise”.

(3) Figure 6: This experiment confuses me. The authors purport to fractionate proteins using differential lysis, but the proteins they detect are supposed to be transmembrane proteins and thus should always be found associated with the pellet, whether lysis is done using equinatoxin or saponin. Have they discovered a currently unknown trafficking pathway to tell us about? Whilst there is a lot of discussion about the trafficking pathways for TM proteins through the host cell, a number of studies have shown that these proteins are generally found in a membrane-bound state. The authors should elaborate, or choose an experiment that is capable of showing compartment-specific localisation of membrane-bound proteins (protease protection, for example).

We do not believe we identified a novel trafficking pathway, but that we capture trafficking intermediates of PfEMP1 between the PVM and the RBC periphery, in either small vesicles, and possibly including Maurer’s clefts. These would still be membrane embedded, but because of their small size, not be pelleted using the centrifugation speeds in our study (we did not use ultracentrifugation). This explanation, we believe, is in line with the current hypothesis of PfEMP1 and other exported TMD protein trafficking to the periphery or the Maurer’s clefts.

(4) The red blood cell contains, in addition to HSP70-X, a number of human HSPs (HSP70 and HSP90 are significant in this current case). As the name suggests, these proteins non-specifically shield exposed hydrophobic domains revealed upon partial protein unfolding following thermal insult. I would thus have expected to find significantly more enrichment following heat shock, but this is not the case. Is it possible that the physiological heat shock conditions used in this current study are not high enough to cause a real heat shock?

As noted by the reviewer, we do not see enrichment of red blood cell heat shock proteins following heat stress, either with FIKK10.2-TurboID or in the phosphoproteome. We used a physiologically relevant heat stress that significantly modifies the iRBC, as shown by our functional assays. While a higher temperature might induce an association of red blood cell heat shock proteins, such conditions may not accurately reflect the most commonly found in the context of malaria infection.

**Reviewer #3 (Public review):**
Summary:In this paper, it is established that high fever-like 39 C temperatures cause parasite-infected red blood cells to become stickier. It is thought that high temperatures might help the spleen to destroy parasite-infected cells, and they become stickier in order to remain trapped in blood vessels, so they stop passing through the spleen.Strengths:The strength of this research is that it shows that fever-like temperatures can cause parasite-infected red blood cells to stick to surfaces designed to mimic the walls of small blood vessels. In a natural infection, this would cause parasite-infected red blood cells to stop circulating through the spleen, where the parasites would be destroyed by the immune system. It is thought that fevers could lead to infected red blood cells becoming stiffer and therefore more easily destroyed in the spleen. Parasites respond to fevers by making their red blood cells stickier, so they stop flowing around the body and into the spleen. The experiments here prove that fever temperatures increase the export of Velcro-like sticky proteins onto the surface of the infected red blood cells and are very thorough and convincing.Weaknesses:A minor weakness of the paper is that the effects of fever on the stiffness of infected red blood cells were not measured. This can be easily done in the laboratory by measuring how the passage of infected red blood cells through a bed of tiny metal balls is delayed under fever-like temperatures.

Previous work by Marinkovic et al. (cited in this manuscript) reported that all RBCs, both infected and uninfected, increase in stiffness at 41 °C compared with 37 °C, with trophozoites and schizonts exhibiting a particularly pronounced increase. We agree that it would be interesting to determine whether similar changes occur at physiological fever-like temperatures, and whether this increase in stiffness coincides with the period of elevated protein trafficking. However, here we focused on enhanced protein export using multiple complementary approaches, and have chosen to address rigidity questions in a different study.

**Recommendations for the authors:**

**Reviewer #1 (Recommendations for the authors):**
As mentioned above, a second time point in many of the assays (for example, 36 hrs or later) would be useful to determine if heat stress simply accelerates trafficking of proteins to the RBC or if instead it results in an overall increase in trafficking.

As mentioned earlier: We did not assess cytoadhesion at later stages, but in the supplementary figures we show that at 40 hours post infection both heat stress and control conditions have comparable proportions of VAR2CSA-positive iRBCs. This is true for the DMSO (control wildtype resembling) HA-tagged lines of HSP70x and PF3D7_072500 (Supplementary Figures 9 and 12 respectively). The end level of VAR2CSA is the same in both conditions, but at 24 hours post infection it is higher following heat stress, indicating that trafficking is accelerated.

In the text, the authors frequently mention changes in the parasites' phenotype in response to heat stress; however, the way it is described is a bit ambiguous and can be confusing. For example, on page 3, they state that "Following heat stress, significantly more iRBCs (57.6% +/-19.4%) cytoadhered.....". From this sentence, it is not initially clear if the end result is cytoadherence of 57.6% of iRBCs or if this refers to an increase of 57.6%. This could be stated explicitly (e.g., "an increase of 57.6% +/- 19.4%") to avoid confusion. Similar descriptions of the results are found throughout the paper.

We agree this is confusing and altered the text accordingly.

The authors might consider citing and discussing the paper from Andrade et al (Nat Med, 2020, 26:1929-1940), which describes longer circulation times (less cytoadherence) by parasites in the dry season (asymptomatic patients) than in febrile patients in the wet season (stronger cytoadhesion of younger stages). This would seem to be consistent with the data presented here.

We are aware of the Andrade study, but chose not to cite it in this context since the reported differences in cytoadhesion appear more consistent with PfEMP1 expression levels, as hypothesized by the authors, than with altered trafficking.

**Reviewer #2 (Recommendations for the authors):**
General comments on the text:(1) "Approximately 10% of the proteins encoded by *P. falciparum* are predicted to be exported beyond the parasite plasma membrane (PPM) into the parasitophorous vacuole lumen (PVL) and subsequently across the parasitophorous vacuole membrane (PVM) into the RBC cytosol."To my knowledge, it has not been really demonstrated that all exported proteins take this route (transfer step in the PVL), and how transmembrane proteins transfer from the parasite to the erythrocyte is still poorly understood. I recommend that the authors rephrase this for precision.

We agree with this reviewer and will change the statement.

Changes:

We have clarified these statements to accurately reflect the current understanding of protein export. Approximately 10% of *P. falciparum* encoded proteins are predicted to be exported beyond the parasite plasma membrane, with many thought to pass through the parasitophorous vacuole lumen (PVL) and parasitophorous vacuole membrane (PVM) into the RBC cytosol, although the exact routes for transmembrane proteins are not fully understood.”

(2) "Charnaud et al. 25, but not Cobb et al. 26, found HSP70x to be essential for normal PfEMP1 trafficking, although both studies concluded that HSP70x is dispensable for intraerythrocytic parasite growth at 37 {degree sign}C."The trafficking block in Charnaud is likely due to a delay in parasite development and cannot thus really be directly related to PfEMP1 trafficking.

Charnaud et al., report: “Microscopy of Giemsa stained IE indicated that ΔHsp70-x appeared similar to CS2 with no obvious abnormalities (Fig 2c). To more accurately quantify changes in maturation through the cell cycle, the DNA content of parasites stained with ethidium bromide was measured by flow cytometry (Fig 2d). This indicated that most parasites had the same DNA content at each timepoint and were maturing at the same rate.”

Thus, we cannot conclude that the trafficking phenotype reported in the Charnaud study can be attributed to a growth delay. This is also supported by only minor changes in the transcriptome, which would likely be more widely perturbed if there was a significant growth delay. However, we will change the statement “Charnaud et al., found HSP70x to be essential for normal PfEMP1 trafficking”, to ”…important for PfEMP1 trafficking” to more precisely reflect the data.

(3) "NanoLuciferase (NanoLuc) fusion proteins and compartment-specific isolation confirmed a greater abundance of PfEMP1 in the RBC cytosol following heat stress."Please see my comments about the differentiation between soluble and TM-containing proteins. One would expect that PfEMP1 is membrane-integrated, and thus should not be found in the cytosol (implying a soluble form).

See our response above.

(4) "Importantly, heat stress did not accelerate parasite development through the asexual life cycle (Supplementary Figure 1)."The authors should constrain this statement to the time frame in which the heat-shock was given. Previous publications have shown a speeded-up development only in younger-stage parasites, which the authors did not study.

We will re-phrase.

Changes:

We have rephrased the sentence to clarify the time window of heat stress: ”Importantly, heat stress between 16-24 hours post-invasion did not accelerate parasite development through the asexual life cycle (Supplementary Figure 1).” The supplementary figure title has also been updated to match.

(5) I recommend that the authors include line numbers. This makes the reviewers' lives much easier.

We agree and apologize for this oversight.

We now added line numbers.

**Reviewer #3 (Recommendations for the authors):**
(1) All the experiments have been performed to a very high standard, and I have no major questions about the results. However, the paper would go up to the next level if the effect of fever temperatures on the stiffness of the iRBCs had been investigated by measuring the passage of iRBCs through an artificial spleen where a bed of metal spheres mimics interendothelial splenic slits.

See our comment from above.

(2) With respect to Figures 5E, 6C, and 6E, why was there not a decrease in bioluminescence levels at 39 {degree sign}C for Sap and NP40 to match the increase in EqtII?

The assay is not performed as a sequence of permeabilisation steps. Instead, samples are split into three parallel treatments: one with EqtII, one with Saponin, and one with NP40. The protein measured in each case reflects the total released under that specific condition rather than being cumulative. Therefore, the NP40 fraction includes proteins from the Saponin-accessible compartment, the EqtII-accessible compartment, and the parasite cytosol.

(3) In the Supplementary gene maps, I could not read the white text on the black gene boxes.

We apologize: these have not converted well and will be altered with the revised version.

Changes

We have significantly increased the size of all fonts within the gene maps and improved the resolution of the figures to improve readability.

(4) In Figure S6, why does HSP70-x look different between parts C and D IFAs, with the latter showing much more export?

We agree these IFAs are not optimal and we will provide better images.

New Results:

Immunofluorescence microscopy, including the localisation of the two HA-tagged proteins (PF3D7_1039000 and PF3D7_0702500), has been repeated and higher-quality images are now included in the updated manuscript (Supplementary Figures 9 and 11). These figures now include multiple images of HA-tagged staining to more accurately represent the observed localisation and export patterns.

(5) Would the authors care to comment on what kinase might be additionally phosphorylating at 39 {degree sign}C?

We presume these are Maurer’s clefts FIKK kinases as most of the hyperphosphorylated proteins are MC residents. However, without directly testing for this using conditional KO parasite lines, we cannot exclude that host kinases are also playing a role.

(6) Could the additional assembly of PSAC at the iRBC membrane be important for survival at 39 {degree sign}C?

We have tested to see if nutrient uptake helps parasite survival during heat stress in the presence of furosemide and lower nutrient concentrations, but did not see a difference in growth following heat stress compared to control temperature conditions.

New Results:

We have added a new supplementary figure (Supplementary Figure 4) detailing experiments testing parasite growth under altered nutrient availability using two approaches (sub-lethal furosemide concentrations or reduced-nutrient RPMI) and with or without a 40°C heat stress applied between 16-24 hpi.

The main text now references this data: “Culturing parasites in sub-lethal furosemide concentrations or in reduced nutrient media lead to reduced parasitaemia (Supplementary Figure 4). However, the parasitaemia is not further reduced following heat stress. This shows that increased PSAC levels/activity do not enhance parasite survival under conditions of limited nutrient availability either from furosemide-induced nutrient deprivation or a reduced nutrient media composition.”

These experiments show that nutrient uptake does not improve parasite survival during heat stress compared to control temperature conditions.

(7) Would the authors like to speculate on how higher temperatures increase the transport of exported proteins with TMDs?

There are many possible explanations, one of which is that unfolding of the hydrophobic TMD domains is favoured at elevated temperatures. However, we have no data to support this hypothesis and therefore refrained from particularly stating this possibility.